# Learning to Complement with Multiple Humans

## Abstract

Real-world image classification tasks tend to be complex, where expert labellers are sometimes unsure about the classes present in the images, leading to the issue of learning with noisy labels (LNL). The ill-posedness of the LNL task requires the adoption of strong assumptions or the use of multiple noisy labels per training image, resulting in accurate models that work well in isolation but fail to optimise human-AI collaborative classification (HAI-CC). Unlike such LNL methods, HAI-CC aims to leverage the synergies between human expertise and AI capabilities but requires clean training labels, limiting its real-world applicability. This paper addresses this gap by introducing the innovative Learning to Complement with Multiple Humans (LECOMH) approach. LECOMH is designed to learn from noisy labels without depending on clean labels, simultaneously maximising collaborative accuracy while minimising the cost of human collaboration, measured by the number of human expert annotations required per image. Additionally, new benchmarks featuring multiple noisy labels for both training and testing are proposed to evaluate HAI-CC methods. Through quantitative comparisons on these benchmarks, LECOMH consistently outperforms competitive HAI-CC approaches, human labellers, multi-rater learning, and noisy-label learning methods across various datasets, offering a promising solution for addressing real-world image classification challenges.

## 1 Introduction

When dealing with real-world image classification problems, it is common that labellers often encounter difficulties in accurately labelling images (Carneiro, 2024). This can occur for various reasons, such as the difficulty of the problem (e.g., medical diagnosis (Chen et al., 2023) or fine-grained classification (Wei et al., 2023)) or due to the labeller's lack of experience (Kamar et al., 2012). In this paper, we primarily focus on the challenges posed by the complexity of the classification task, assuming that labellers are experts in classifying dataset images.

Two separate research communities have addressed this challenge by making different assumptions and proposing quite distinct solutions. The learning with noisy-label (LNL) research community focuses on mitigating the presence of noise in the labels with sophisticated training methods (Song et al., 2022; Carneiro, 2024; Ji et al., 2021) that aim to maximise model accuracy during testing when they operate in isolation. Given the ill-posedness of the LNL problem (Liu et al., 2023b), current solutions either need to impose strong assumptions (e.g., clean-label training samples tend to have smaller losses than noisy-label samples) or they rely on multiple noisy labels per training image to build multi-rater learning (MRL) methods (Ji et al., 2021). On the other hand, the human-AI collaborative classification (HAI-CC) community (Dafoe et al., 2021) focuses on the development of methods that assume the presence of clean and multiple noisy training labels to exploit the complementary performance of human experts and AI to produce a collaborative approach that has higher accuracy than both the expert's and the AI's accuracy. However, HAI-CC methods require ground truth labels in the training set, restricting their applicability in real-world scenarios that contain exclusively noisy labels for training. Apart from the need of clean labels, another limitation of HAI-CC methods is that they collaborate only with single users, limiting real-world deployment. There are notable HAI-CC exceptions (Hemmer et al., 2022; Verma et al., 2023) that can learn to complement with or defer to multiple experts, but their reliance on clean-label samples, absence of collaboration cost

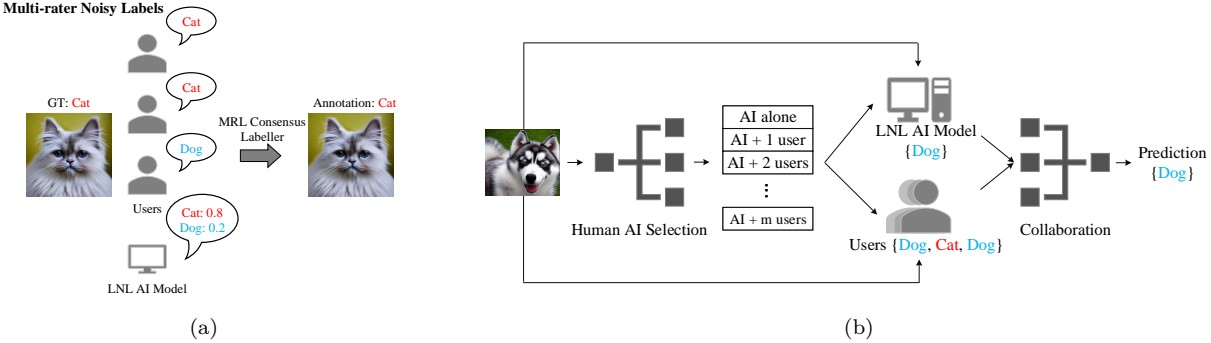

(a)  (b)

Figure 1: LECOMH is the first human-AI collaborative classification (HAI-CC) method that learns exclusively from multiple noisy labels and collaborates with multiple experts. Its primary objective is to optimise HAI-CC accuracy while concurrently minimising collaboration costs, measured by the number of human expert annotations required for image classification. To enable the learning from multiple noisy labels, we first train an AI model using learning with noisy label (LNL) techniques, followed by a multi-rater learning (MRL) to produce a consensus label that is then used as the ground truth label for training the two stages of HAI-CC. The first stage is the *Human-AI Selection Module* that estimates the number of human predictions needed for efficient and accurate human-AI collaborative classification, and the second stage is the *Collaboration Module* that produces the final prediction.

optimisation (Hemmer et al., 2022), and lack of human-AI ensemble classification (Verma et al., 2023) restrict their applicability in real-world scenarios. As a result, it is possible to notice a remarkable research gap, where LNL and MRL methods can handle noisy labels in training, but they do not collaborate with users during testing, while HAI-CC methods collaborate with users during testing, but they rely on clean labels for training and rarely collaborate with multiple experts.

This paper addresses the research gap exposed above with the innovative Learning to Complement with Multiple Humans (LECOMH) approach and the introduction of new HAI-CC benchmarks. LECOMH, shown in Fig. 1, is designed to learn from multiple noisy labels per sample to maximise the HAI-CC accuracy and minimise the multiple user collaboration costs, measured by the number of human expert annotations in the collaborative classification of a test image. The proposed benchmarks assess HAI-CC methods with datasets containing multiple noisy labels in training and testing. Overall, the key contributions of the paper are:

- the first HAI-CC method, referred to as LECOMH, that can be trained exclusively from multiple noisy labels per training image to maximise the collaborative classification accuracy of teams of AI and multiple experts, while minimising the collaboration costs, measured by the number of human experts used in HAI-CC, and

- new benchmarks to assess HAI-CC methods on classification problems containing multiple noisy labels in the training and testing sets, paving the way for a more comprehensive performance evaluation of real-world applications.

The empirical evaluation shows that LECOMH consistently demonstrates superior performance than state-of-the-art (SOTA) HAI-CC methods (Mozannar et al., 2023; Hemmer et al., 2022; Verma et al., 2023) in the newly-introduced benchmarks with higher accuracy for equivalent collaboration costs. Furthermore, LECOMH is the only HAI-CC method in our experiments that outperforms expert labellers and isolated LNL methods across all datasets.

The rest of this work is organised as follows: Section 2 presents a brief review of previous studies on noisy-label learning, multi-rater learning and human-AI collaborative classification. Section 3 describes the proposed Learning to Complement with Multiple Humans approach. Section 4 introduces new benchmarks to assess human-AI collaborative classifiers with multi-rater noisy-label datasets, including new benchmarks in

CIFAR-10, Chaoyang, and NIH datasets. Section 5 presents and analyses the empirical evaluation. Finally, Section 6 summarises and conclude this study.

## 2    Related Work

LECOMH is a new method that jointly addresses the challenges of learning with multiple noisy labels and human-AI collaboration. Thus, in this section, we review relevant studies in learning with noisy labels, multi-rater learning, and human-AI collaborative classification.

### 2.1    Learning with Noisy Labels (LNL)

LNL is a challenging problem that has received increasing attention by the machine learning community (Song et al., 2022; Carneiro, 2024). Since learning with noisy labels is an ill-posed problem, it necessitates the imposition of specific constraints to facilitate the discovery of a viable solution. One common constraint is the small-loss hypothesis (Li et al., 2020; Jiang et al., 2018; Arazo et al., 2019), which posits that clean-label training samples incur smaller losses than noisy-label samples. Another constraint, known as clusterability (Zhu et al., 2021b), assumes that a training sample and its two nearest neighbours share the same clean label.

The development of these constraints allowed the proposal of a vast number of LNL methods, which include: robust loss functions (Zhang & Sabuncu, 2018; Ghosh et al., 2017), co-teaching (Jiang et al., 2018; Han et al., 2018), label cleaning (Yuan et al., 2018; Jaehwan et al., 2019), semi-supervised learning (SSL) (Li et al., 2020; Ortego et al., 2021), iterative label correction (Chen et al., 2021; Arazo et al., 2019), meta-learning (Ren et al., 2018; Zhang et al., 2020; Zhang & Pfister, 2021; Xu et al., 2021), and graphical modelling (Garg et al., 2023). Among these, SSL represents a dominant technique used in LNL (Li et al., 2020). Graphical models also show accurate results for some specific LNL problems (e.g., instance-dependent noise) (Garg et al., 2023; 2025).

Recent research has shown that the standard LNL with single noisy label per sample without any additional constraints is non-identifiable (Liu et al., 2023b; Nguyen et al., 2023), unless additional noisy labels of each sample are available. In other words, multiple noisy labels per training sample are crucial to obtain high-performance models. Such studies also emphasise the importance of multi-rater learning which is discussed in the following subsection.

Although much research has been developed to address the LNL problem, none of the methods above collaborate with human during testing, even though such collaboration has the potential to improve the LNL results (Rastogi et al., 2023).

### 2.2    Multi-rater learning

Multi-rater learning (MRL) is a vast field in machine learning that deals with imperfect, noisy or incomplete label datasets. MRL aims to train models by aggregating multiple noisy labels annotated by multiple humans to "reliable ground truth" or *consensus labels*. Early MRL methods focus on majority voting (Zhou, 2012), or probabilistic modelling (e.g., ground truth as a latent random variable) and apply the Expectation - Maximisation algorithm to estimate the ground truth (Dawid & Skene, 1979; Whitehill et al., 2009; Raykar et al., 2010; Rodrigues et al., 2014; 2017). Another MRL approach is to investigate the variability of annotators to improve the reliability of the ground truth estimation. Specifically, the inter-observer variability methods focus on characterising the disagreement between annotators (Raykar et al., 2009; Guan et al., 2018; Mirikharaji et al., 2021; Ji et al., 2021), while the intra-observer strategy studies the inconsistency in the labelling pattern of a specific annotator (e.g., through confusion matrices) (Khetan et al., 2017; Tanno et al., 2019; Wu et al., 2022a). A joint learning method combining both the inter- and intra-annotator strategies is also proposed to integrate the strengths of both approaches (Wu et al., 2022a). One drawback in those studies is the assumption of sample-independence, potentially deviating from real-world applications where annotation error might depend on both samples and annotators. Such a challenge motivates the study by Gao et al. (2022) to learn a sample-dependent model. Recently, UnionNet (Wei et al., 2022) has been developed

to integrate labelling information from all annotators, leveraging this collective input to better coordinate responses across multiple contributors. CrowdAR (Cao et al., 2023) focuses on predicting the reliability of annotations to directly assess the quality of crowd-sourced data, which is used as a soft annotation to produce a consensus training label. ADMoE (Zhao et al., 2023) adopts a Mixture of Experts (MoE) architecture to foster specialised and scalable learning from multiple noisy sources, particularly targeting anomaly detection. Furthermore, GeoCrowdNet (Ibrahim et al., 2023) introduces two identifiability-enhanced approaches for end-to-end crowdsourcing, while BayesianIDNT (Guo et al., 2023) approximates instance-dependent noise transition matrices using a Bayesian network with a hierarchical spike and slab prior. Zhang et al. (2024) tackle the challenge of annotation sparsity, where annotators provide only a limited number of labels. They propose a meta-learning-based coupled confusion correction method to refine confusion matrices derived from two networks. CROWDLAB (Goh et al., 2022) is a state-of-the-art (SOTA) MRL method that produces consensus labels using a combination of multiple noisy labels and the predictions by an external classifier.

Despite being widely studied, most MRL approaches do not integrate SOTA LNL methods nor introduce a collaboration mechanism with humans during testing. Addressing these two issues has the potential to improve the accuracy of MRL methods, as we demonstrate in this paper.

An important distinction needs to be made at this stage is crowd-sourcing which largely overlaps with multi-rater learning, and is often used interchangeably in the literature. Here, we refer crowd-sourcing as a paradigm that quantifies the quality or the behaviour of labellers where the end goal is to optimise annotation processes through either data assignment or labeller reward (Vaughan, 2018). In contrast, MRL approaches aim to infer ground truth labels from multiple noisy annotations which are most likely collected via crowd-sourcing.

### 2.3 Human-AI Collaborative Classification (HAI-CC)

In machine learning, the vast majority of AI systems have been optimised in isolation, without considering the implications of human-AI collaborative classification (Rosenfeld et al., 2018; Serre, 2019; Kamar et al., 2012). Scientifically, the isolated development of AI systems is correct, but in practice, the influence of AI decisions on humans is unpredictable and represents a critical point to study. Recently, Chiou & Lee (2023) studied how AI decisions influence human experts, reaching the conclusion that the trustworthiness of AI depends on both model confidence (Lu & Yin, 2021; Yin et al., 2019) and explainability (Shin, 2021; Weitz et al., 2019). Nevertheless, this is a two-way lane, and while it is important to consider how AI influences humans, we must also consider that humans can affect AI decision, which is the main study topic of HAI-CC (Bansal et al., 2021; Agarwal et al., 2023; Vodrahalli et al., 2022; Wu et al., 2022b; Wilder et al., 2021). In general, when classifying an image, there are three options to be considered by HAI-CC systems:

    I ) *AI predicts alone*;

   II ) *AI defers decisions to human experts*;

  III ) *AI makes a joint decision with human experts*.

Below, we explain the main HAI-CC approaches being studied, relating them to these three options.

#### 2.3.1 Learning to Defer (L2D)

L2D methods focus on HAI-CC options (I) and (II) above, meaning that the decision relies on a "single" prediction made by the AI model or the human expert. L2D aims to learn a classifier and a rejector to decide when a human expert prediction should replace the AI prediction (Madras et al., 2018; Keswani et al., 2021; Narasimhan et al., 2022; Mao et al., 2023). This *rejection learning* (Cortes et al., 2016) approach was generalised by considering the human expertise in the decision-making process (Madras et al., 2018). Further investigation into L2D methods then concentrated on the development of different surrogate loss functions that are consistent with the Bayes-optimal classifier obtained in the case of 0-1 loss (Narasimhan et al., 2022; Charoenphakdee et al., 2021; Raghu et al., 2019; Okati et al., 2021; Mozannar & Sontag, 2020; Verma & Nalisnick, 2022; Mozannar et al., 2023; Charusaie et al., 2022; Cao et al., 2024; Straitouri et al., 2023; Liu

et al., 2024; Mozannar et al., 2022). Another L2D approach is the score-based triages (Raghu et al., 2019) which introduced two error prediction algorithms for human and machine errors to optimise the decision and reduce overall error. This was extended by the differentiable triages that utilise a deterministic threshold rule for triage decisions, where the threshold is derived from the differences in errors between the model and human decisions on individual instances (Okati et al., 2021). One common limitation of L2D-based methods is the reliance on the single-expert setting, overlooking the more complex environments with the availability of multiple experts. Hence, recent research in L2D has shifted the focus to the multiple-expert setting (Verma et al., 2022; Mao et al., 2023; Verma et al., 2023; Keswani et al., 2021; Babbar et al., 2022; Mao et al., 2024; Hemmer et al., 2023; Tailor et al., 2024; Leitão et al., 2022). Despite such an extensive research, current L2D-based learning methods have not been designed to enable AI models and human experts to jointly produce a final classification. To address this gap, "learning to complement" methods have been introduced as explained below.

### 2.3.2 Learning to Complement (L2C)

L2C methods focus on HAI-CC options (I) and (III), aiming to optimise the collaboration between human experts and the AI model to maximise the expected utility of the human-AI decision (Wilder et al., 2021; Steyvers et al., 2022; Kerrigan et al., 2021; Liu et al., 2023a; Bansal et al., 2021; Hemmer et al., 2022; Charusaie et al., 2024). Kerrigan et al. (2021) proposed to combine human and model predictions via confusion matrices and model calibration. Steyvers et al. (2022) introduced a Bayesian framework for combining the predictions and different types of confidence scores from humans and machines, demonstrating that a hybrid combination of human and machine predictions leads to better performance than combinations of human or machine predictions alone. Recently, Liu et al. (2023a) leverages perceptual differences between humans and AI to make a human-AI system outperform humans or AI alone, while Hemmer et al. (2022) introduced a model featuring an ensemble prediction involving both AI and human predictions, yet it does not optimise the collaboration cost. Recently, Charusaie et al. (2024) introduced a method that determines whether the AI model or the user should predict independently, or if they should collaborate on a joint prediction, but it overlooks the cost of such collaboration.

One common assumption of existing HAI-CC methods, including both L2D- and L2C-based approaches, is the availability of *clean* labels. This is, however, impractical because real-world settings typically only contain multiple noisy labels per sample. Furthermore, although collaborative classification has been explored (Hemmer et al., 2022), the cost produced by such a collaboration is not taken into account. In contrast, our proposed LECOMH is designed to work with multiple noisy labels per sample in the training set by exploring LNL and MRL methods. Furthermore, LECOMH jointly maximises human-AI classification accuracy and minimises the collaboration cost, measured by the number of human experts annotations used to classify an image.

## 3 Learning to Complement with Multiple Humans (LECOMH)

Let $\mathcal{D} = \{(x_i, \mathcal{M}_i)\}_{i=1}^{|\mathcal{D}|}$ be the noisy-label multi-rater training set, where $x_i \in \mathcal{X} \subset \mathbb{R}^d$ denotes a sample as a $d$-dimensional vector, and $\mathcal{M}_i = \{m_{i,j}\}_{j=1}^{M}$ denotes the noisy annotations of $M$ human experts for the sample indexed by $i$, with $m_{i,j} \in \mathcal{Y} \subseteq \{0,1\}^{|\mathcal{Y}|}$ being a one-hot label.

Our methodology, as layout in Fig. 1b, consists of: 1) an *AI Model* pre-trained with LNL techniques to enable the production of a training sample consensus label by the multi-rater learning approach CROWDLAB (Goh et al., 2022), 2) a *Human-AI Selection Module* that predicts the collaboration format (i.e., AI alone, AI + 1 user, AI + 2 users, etc.), and 3) a *Collaboration Module* that aggregates the predictions selected by the Human-AI Selection Module to produce a final prediction. Note that our training and testing, as explained below, are designed to be unbiased to any specific labeller, so when we select AI + 1 user or AI + 2 users, the users are randomly selected from our pool of users.

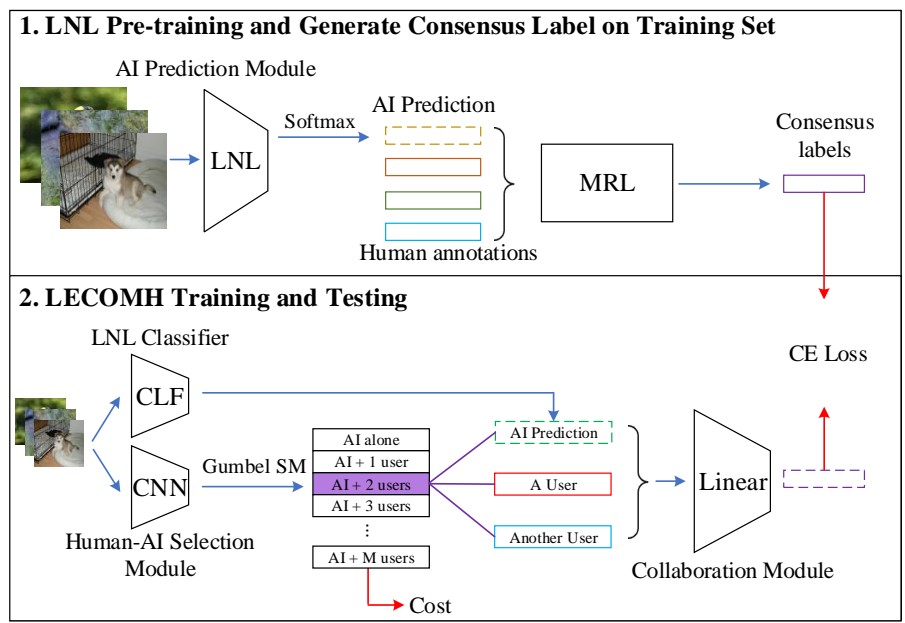

Figure 2: The proposed LECOMH consists of two main steps: 1. *(top)* estimate the consensus labels by exploiting a pre-trained LNL model coupled with an MRL module (Goh et al., 2022); and 2. *(bottom)* train an LNL classifier (CLF) and a human-AI selection module by minimising both the classification error and the collaboration cost. In particular, the training step involves: 1) building the set of AI predictions and user labels, 2) training the Human-AI Selection Module to estimate the number of users to collaborate with the AI classifier, and 3) training the Collaboration Module to produce a final classification using AI predictions and selected users' labels. Testing involves similar steps to generate the final prediction.

## 3.1 Training

LECOMH maximises classification accuracy and minimises collaboration costs in a human-AI collaborative classification setting, where cost is proportional to the number of users that are asked to provide labels. Our training has three phases (see Fig. 2): 1) pre-training the LNL AI Model, 2) generating consensus labels for the training set using the multi-rater learning CROWDLAB method (Goh et al., 2022), and 3) training of LECOMH's Human-AI Selection and Collaboration Modules. We provide more details as follows:

### 3.1.1 LNL Pre-training and Consensus Label Generation

We use SOTA LNL techniques (Wang et al., 2023; Garg et al., 2023; Zhu et al., 2021a; Liu et al., 2022) to train the LNL AI model $f_\theta : \mathcal{X} \to \Delta^{|\mathcal{Y}|-1}$, where $\Delta^{|\mathcal{Y}|-1}$ denotes the $|\mathcal{Y}|$-dimensional probability simplex, and $\theta \in \Theta$ is the classifier's parameter. This LNL training uses the training set, where the noisy label of a sample $x_i$ is randomly selected as one of the experts' annotations in $\mathcal{M}_i$. For the consensus label generation, we leverage the SOTA multi-rater learning method CROWDLAB (Goh et al., 2022) that takes the training samples and experts' labels $(x_i, \mathcal{M}_i) \in \mathcal{D}$, together with the AI classifier's predictions $\hat{y}_i = f_\theta(\mathbf{x}_i)$ for each sample in $\mathcal{D}$ to produce a consensus label $\hat{y}_i^c \in \mathcal{Y}$ and a quality (or confidence) score $\alpha$. Formally, the formation of the consensus label dataset can be written as:

$$\mathcal{D}^c = \{(x_i, \hat{y}_i^c, \mathcal{M}_i)|(x_i, \mathcal{M}_i) \in \mathcal{D} \land (\hat{y}_i, \alpha_i) = \mathsf{CROWDLAB}(x_i, f_\theta(x_i), \mathcal{M}_i) \land \alpha_i > 0.5\}, \tag{1}$$

which is used by the LECOMH training, as explained below. We use CROWDLAB for MRL because it can combine labels from annotators and the pre-trained LNL AI model to produce highly accurate consensus labels (Goh et al., 2022) for the subsequent LECOMH training.

### 3.1.2   LECOMH training

The proposed LECOMH comprises the Human-AI Selection Module and the Collaboration Module, as shown in Fig. 2. The Human-AI Selection Module, represented by $g_\phi : \mathcal{X} \to \Delta^M$, predicts a probability of having either an isolated AI prediction (1st dimension) or a combined prediction between AI and multiple users (remaining $M$ dimensions). In other words, the $j$-th index of $g_\phi^{(j)}(x)$ represents selecting the AI model and $j - 1$ annotators. The Collaboration Module, represented by $h_\psi : \left(\Delta^{|\mathcal{Y}|-1}\right)^{M+1} \to \Delta^{|\mathcal{Y}|-1}$, takes the AI prediction in the first input, and the remaining user predictions selected by $g_\phi(.)$ to produce the final classification prediction $\tilde{y}_i$ defined as follows:

$$\tilde{y}_i = h_\psi \left( \mathsf{p} \left( g_\phi(x_i), f_\theta(x_i), \mathsf{rand}(\mathcal{M}_i) \right) \right), \tag{2}$$

where:

$$\mathsf{p} \left( g_\phi(x), f_\theta(x), \mathsf{rand}(\mathcal{M}) \right) = \begin{cases} \begin{bmatrix} f_\theta(x) & \mathbf{0}_{|\mathcal{Y}|} & \dots & \mathbf{0}_{|\mathcal{Y}|} \end{bmatrix}^\top & \text{if } \max_j g_\phi^{(j)}(x) = g_\phi^{(1)}(x) \\ \begin{bmatrix} f_\theta(x) & m_{i,1} & \dots & \mathbf{0}_{|\mathcal{Y}|} \end{bmatrix}^\top & \text{if } \max_j g_\phi^{(j)}(x) = g_\phi^{(2)}(x) \\ \dots \\ \begin{bmatrix} f_\theta(x) & m_{i,1} & \dots & m_{i,M} \end{bmatrix}^\top & \text{if } \max_j g_\phi^{(j)}(x) = g_\phi^{(M+1)}(x), \end{cases} \tag{3}$$

with $g_\phi^{(j)}(.)$ denoting the $j$-th output from the Human-AI Selection Module and $\mathsf{rand}(\mathcal{M})$ representing a function that randomly selects the experts' annotations to avoid bias toward any specific experts' annotations.

The Human-AI Selection Module and the Collaboration Module is trained by minimising the cross-entropy loss $\ell(.,.)$ between the consensus label $y_i^c$ and the final prediction $\tilde{y}_i$, plus an additional term that regularises the cost as follows:

$$\min_{\phi,\psi} \frac{1}{|\mathcal{D}^c|} \sum_{(x_i, \hat{y}_i^c, \mathcal{M}_i) \in \mathcal{D}^c} \ell\left(\hat{y}_i^c, \tilde{y}_i\right) + \lambda \times \mathsf{cost}(g_\phi(x_i)), \tag{4}$$

where $\tilde{y}_i$ is the consensus label of sample $x_i$, defined in Eq. (2), $\lambda$ is a hyper-parameter that weights the cost function, and

$$\mathsf{cost}(g_\phi(x)) = \sum_{j=1}^{M+1} g_\phi^{(j)}(x) \times (j-1). \tag{5}$$

For the cost in Eq. (5), when the AI model is selected to predict alone, the selection module would output a probability such that $\max_j g_\phi^{(j)}(x) = g_\phi^{(1)}(x)$, resulting in $\mathsf{cost}(g_\phi(x)) = 0$. Thus, for the case that $\max_j g_\phi^{(j)}(x) = g_\phi^{(K)}(x)$ for $K \in [2, M]$, then $\mathsf{cost}(g_\phi(x)) \approx K - 1$. In other words, the cost in Eq. (5) represents the cost of one unit per expert's annotation.

As explained above and depicted in Fig. 2 *(bottom)*, the human-AI training has two main steps: selecting the collaboration format (i.e., AI and the number of experts) and making the prediction through the collaboration module. The selection of the collaboration format is estimated by sampling from the probability vector output of the selection module $g_\phi(x)$. Naively sampling from such a categorical distribution is non-differentiable, prohibiting the training using stochastic gradient descent. To avoid that, we employ the concrete distribution (Maddison et al., 2017), also known as Gumbel-softmax trick (Jang et al., 2017), to approximate such sampling, making it trainable with SGD.

### 3.2   Testing

Testing starts from the LNL AI prediction, followed by the human-AI selection module prediction of the categorical distribution of the probability of the AI model running alone or collaborating with a set of $K \in [1, M]$ users, resulting in a cost of $K$. Note that the human-AI selection module's behavior poses a challenge in determining the number of users to collaborate with the AI classifier, an argmax operation on the module's output tends to favor the AI model alone. Therefore, after deciding on the number of users

to collaborate, using Gumbel-softmax on $g_\phi(x)$, we randomly select testing users, and concatenate their predictions with the AI prediction to serve as input to the collaboration module, which outputs the final classification, following the procedure defined in Eqs. (2) and (3).

### 3.3 Theoretical Analysis of the Impact of Label Noise to Consensus Label Quality

Assume that we have $n$ annotators, one of them being the AI model and the rest representing the human annotators. Each annotator has a probability $p_i$ of providing the correct label, where $p_i > 0.5$ (i.e., better than random chance). Representing the true label with $y^*$ and the $i^{th}$ annotator's label as $y_i$, we have $p_i = \Pr(y_i = y^*)$. We assume that the consensus labelling follows the majority voting rule, where the consensus label is determined by majority voting among annotators. Let these $n$ annotators label a training image, where the consensus label is represented by $\hat{y}$. Such consensus label is correct with the following probability:

$$\Pr(\hat{y} = y^*) = \Pr\left(\sum_{i=1}^{n} \delta(y_i = y^*) > \frac{n}{2}\right), \tag{6}$$

where $\delta(y_i = y^*)$ is equal to one if $y_i = y^*$, and zero otherwise. Since $\sum_{i=1}^{n} \delta(y_i = y^*)$ follows a Binomial distribution, Eq. (6) can be simplified to:

$$\Pr(\hat{y} = y^*) = \sum_{k=\lceil n/2 \rceil}^{n} \binom{n}{k} p^k (1-p)^{n-k}, \tag{7}$$

Using the Central Limit Theorem, as $n \to \infty$, we can approximate the Binomial distribution with the Normal distribution: $\mathsf{Bin}(n, p) = \mathcal{N}(np, np(1-p))$, allowing us to simplify Eq. (7) with:

$$\Pr(\hat{y} = y^*) = \Pr\left(Z > \frac{n}{2} - np\right), \tag{8}$$

where $Z \sim \mathcal{N}(np, np(1-p))$. Note in Eq. (7) that as $n \to \infty$ and $p > 0.5$, the term $\frac{n}{2} - np$ becomes increasingly negative, resulting in $\Pr(\hat{y} = y^*) \to 1$.

Fig. 3a illustrates an example of the consensus accuracy with three experts under varying noise levels, assuming all three experts have the same performance. As the noise rate decreases, the accuracy of the consensus label improves significantly, highlighting the convergence of majority voting in scenarios with lower noise. Furthermore, we present an example of the accuracy-cost trade-off in Fig. 3b, demonstrating the effect of increasing the number of experts, each with 80% accuracy. As the number of experts increases, the consensus accuracy improves and eventually converges to 100%.

## 4 Human-AI Collaborative Benchmarks

### 4.1 New CIFAR-10 Benchmarks

We introduce two new benchmarks with CIFAR-10 (Krizhevsky & Hinton, 2009), which has 50K training images and 10K testing images of size $32 \times 32$. The *first benchmark relies on annotations produced by people* to produce CIFAR-10N (Wei et al., 2021) for training and CIFAR-10H (Peterson et al., 2019) for testing. CIFAR-10N (Wei et al., 2021) has three noisy labels for each CIFAR-10 training image, while CIFAR-10H (Peterson et al., 2019) provides approximately 51 noisy labels per CIFAR-10 testing image. Due to the limitation of three labels per sample in CIFAR-10N, the testing process allows collaboration with at most three users randomly sampled from the pool of users in CIFAR-10H. The *second benchmark*, named multi-rater CIFAR10-IDN (Xia et al., 2022), *is based on synthesised annotations* for training and testing with multi-rater instance-dependent noise. The label noise rates 0.2 and 0.5 for both training and testing sets, with three distinct noisy labels generated for each noise rate to simulate varying human predictions with similar error rates. We also present an experiment involving users with varying noisy label rates. To achieve this, we simulate three synthetic experts with different noise levels, characterized by IDN noise rates of 20%, 30%, and 50% (denoted as IDN{20, 30, 50}). This setup allows us to evaluate the performance of the methods under these conditions.

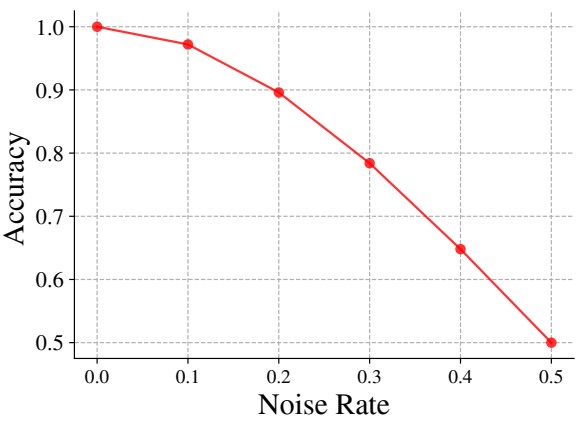

(a) Acc vs. Noise Rate for 3 experts aggregation.

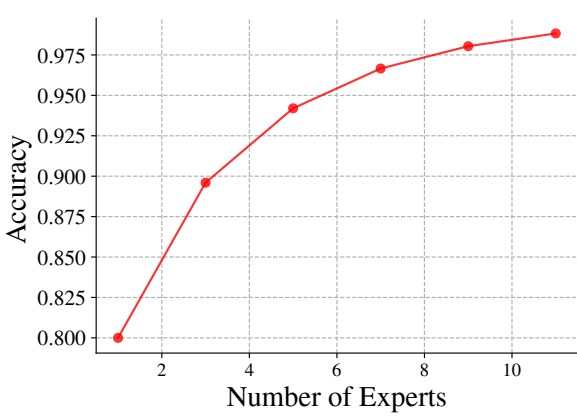

(b) Acc vs. № experts with 80% individual accuracy.

Figure 3: Theoretical analysis of the accuracy vs. noise rate for 3 experts with the same performance, and accuracy vs. number of experts with 80% individual accuracy.

### 4.2 New Chaoyang Benchmark

The Chaoyang dataset has 6,160 colon slides represented as patches of size $512 \times 512$ (Zhu et al., 2021a), where each patch has *three noisy labels produced by real pathologists*. Originally, the dataset had a training set with 4,021 patches for training and 2,139 patches for testing. The training patches had multi-rater noisy labels, while testing patches only contained a unanimous expert agreement on a single label. To create a new benchmark, the dataset was restructured to ensure both training and testing sets contained multiple noisy labels. The entire dataset was reshuffled, resulting in a partition of 4,725 patches for training and 1,435 patches for testing. In this new partition, both sets have multi-rater noisy label patches. The training set comprises 862 patches with 2 out of 3 consensus labels and 3,862 patches with 3 out of 3 consensus labels. The testing set includes 449 patches with 2 out of 3 consensus labels and 986 patches with 3 out of 3 consensus labels. Importantly, patches from the same slide do not appear in both the training and testing sets. The prediction accuracy of the 3 users in the dataset is approximately 93%, 88%, 99% in training and 88.7%, 86.9%, 99% in testing.

### 4.3 Multi-rater NIH Dataset

The multi-rater NIH Chest X-ray dataset (Majkowska et al., 2020; Wang et al., 2017) contains an average of 3 manual labels per image for four radiographic findings on 4,374 chest X-ray images (Majkowska et al., 2020) from the ChestX-ray8 dataset (Wang et al., 2017). We focus on the occurrence of the following clinically important findings: airspace opacity (NIH-AO) and nodule or mass (NIH-NM). The prevalence of NIH-AO and NIH-NM findings are close to 50% and 14%. We selected a total of 2,412 images in the validation set for training and 1,962 images in the testing set for testing. The prediction accuracy of the three users in the NIH-AO dataset is approximately 89%, 94%, 80% in training and 89%, 94%, 80% in testing, while in the NIH-NM dataset, the prediction accuracy of the three users are 92%, 92%, 93% in training and 89%, 90%, 91% in testing.

## 5 Experiments

In this section, we first present the models used in the experiments, then we explain the training and evaluation details, followed by an introduction of the baseline methods. We then show the results and the ablation studies.

### 5.1 Models

All methods are implemented in PyTorch (Paszke et al., 2019) and run on NVIDIA RTX A6000. LNL with random label or majority voting produce similar results, so we decided to use random label since it is simpler. For experiments performed on CIFAR-10N and CIFAR-10H datasets, we employ ProMix (Wang et al., 2023) to pre-train two PreAct-ResNet-18 as the LNL AI models using the set of *Rand1* annotations in CIFAR-10N (Wei et al., 2021). For the experiments of multi-rater learning performed on CIFAR10-IDN, we use InstanceGM (Garg et al., 2023) to pre-train two PreAct-ResNet-18 as the LNL AI models. For the Chaoyang dataset, we follow the practice in NSHE (Zhu et al., 2021a) to pre-train two ResNet-34 using the set of *label_A* annotations. The network having the highest performance is selected for the LNL AI model. For the NIH datasets, we follow the NVUM model (Liu et al., 2022) by pre-training on the ChestXray dataset (Wang et al., 2017) and then fine-tuning on the airspace opacity and nodule and mass classification tasks. All the above models are selected according to their SOTA performance in the respective datasets. The same pre-trained backbones are also used for the Human-AI Selection Module. The Collaboration Module is designated as a two-layer multi-layer perceptron, where each hidden layer has 512 neutrons activated by the Rectified Linear Unit (ReLU) function.

We also measure the performance of those pre-trained models as references. The pre-trained ProMix on CIFAR-10N reaches 97.41% accuracy on the CIFAR-10 test set. In the multi-rater setting relying on CIFAR-10 IDN, the pre-trained InstanceGM reaches an accuracy of 96.64% and 95.90% for the label noise rates at 0.2 and 0.5, respectively. The pre-trained NSHE achieves 82.44% prediction accuracy on Chaoyang, while the pre-trained NVUM is 86.65% and 87.41% on airspace opacity and nodule or mass findings, respectively.

### 5.2 Training and Evaluation Details

For each dataset, the proposed human-AI system is trained for 200 epochs using SGD with a momentum of 0.9 and a weight decay of $5 \times 10^{-4}$. The batch size used is 256 for CIFAR, 96 for Chaoyang and 32 for NIH. The initial learning rate is set at 0.05 and decayed through a cosine annealing. The temperature parameter of the Gumbel-softmax sampling is set at 5. Also, to ensure consistent data range between the LNL classifier and users' predictions, the LNL classifier predictions are normalised with a softmax activation before it is concatenated with the users' annotations. Given that we do not want to bias the performance of the system to any particular user, the users in each collaboration format (e.g., the $m$ users in the format of AI model plus $m$ users) are randomly selected during training and testing for LECOMH. For training, the ground truth labels are set as the consensus labels obtained via CROWDLAB. For testing, the ground truth label is either available from the dataset (e.g., for the CIFAR benchmarks) or from the consensus label obtained from majority voting (e.g., for Chaoyang and Multi-rater NIH benchmarks). The evaluation is based on the prediction accuracy as a function of coverage evaluated on the test sets. Coverage denotes the percentage of examples classified by the AI model, with 100% coverage representing a classification performed exclusively by the classifier, while 0% coverage denoting a classification performed exclusively by the users. To obtain different levels of coverage for LECOMH, we adjust the hyper-parameter $\lambda$ in Eq. (4) during training, where the higher the hyper-parameter $\lambda$, the more emphasis on the cost, and hence, the lower the coverage. To report the mean and standard error of the system accuracy, each run is repeated for 5 trials with different random seeds.

### 5.3 Baselines

We follow (Mozannar et al., 2023) and evaluate LECOMH in the single expert human-AI collaborative classification (SEHAI-CC) setting. We also consider several SOTA methods, such as cross-entropy surrogate (CE) (Mozannar & Sontag, 2020), one-vs-all-based surrogate (OvA) (Verma & Nalisnick, 2022), the confidence method (CC) (Raghu et al., 2019), differentiable triage (DIFT) (Okati et al., 2021), mixture of experts (MoE) (Madras et al., 2018), Realizable Surrogate (RS) (Mozannar et al., 2023), Defer and Fusion (DaF) (Charusaie et al., 2024) as baselines. For a fair comparison, we randomly sample a single annotation for each image as a way to simulate a single expert from the human annotation pools to train those SEHAI-CC methods. To evaluate the accuracy of the system when expect deferral probability changes, we follow previous studies (Mozannar et al., 2023; Mozannar & Sontag, 2020) and plot accuracy-coverage curves,

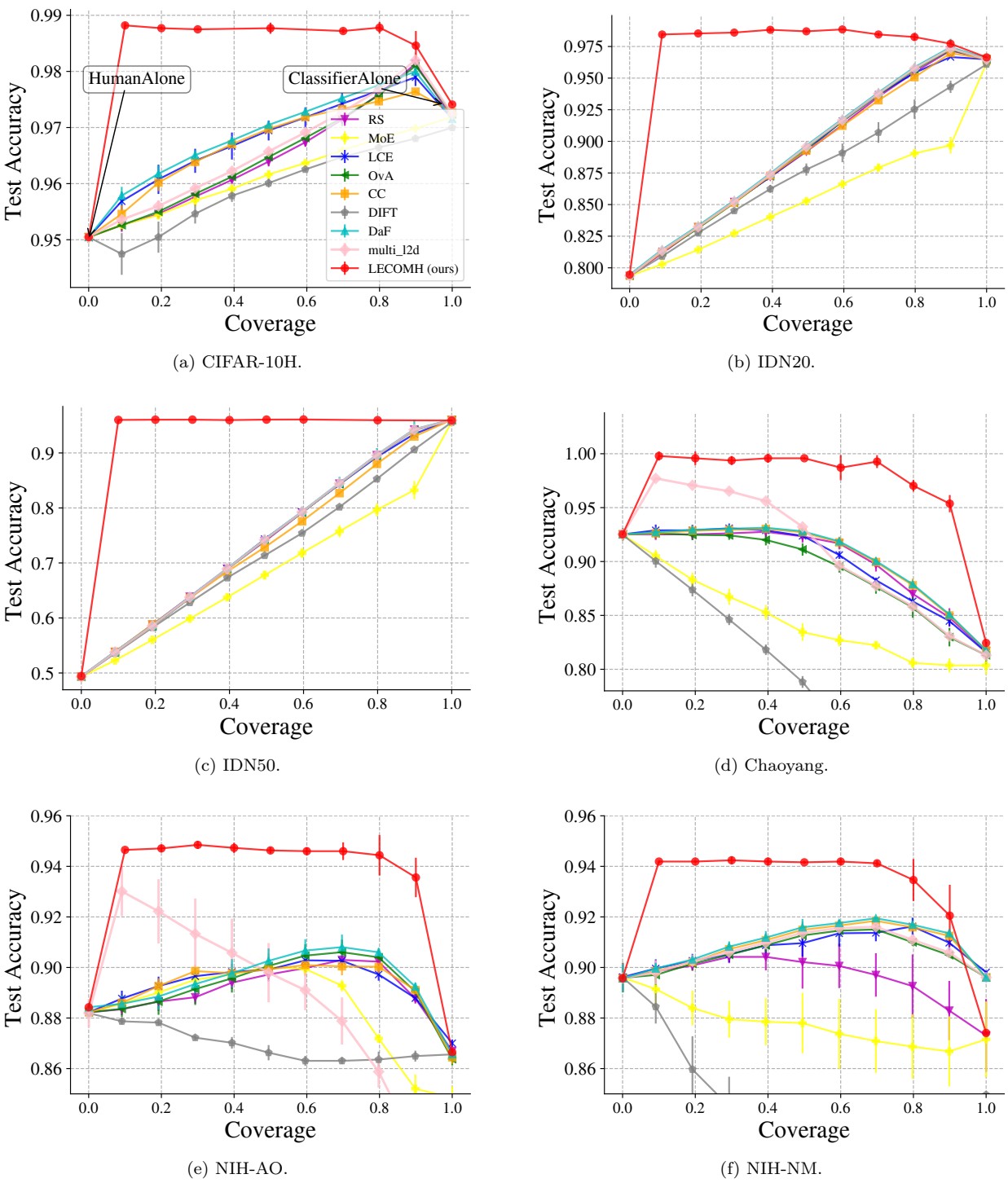

Figure 4: Test accuracy vs. coverage of LECOMH (Ours) and competing SEHAI-CC (Mozannar et al., 2023) and MEHAI-CC (Hemmer et al., 2022; Verma et al., 2023) methods. The SEHAI-CC methods are always pre-trained with LNL techniques, with the single user being simulated with aggregation (majority voting) from the pool of three annotators. Multi_L2D can defer to one of many experts, so we select the label corresponding to the maximum probability of 3 users for each sample to draw the curve.

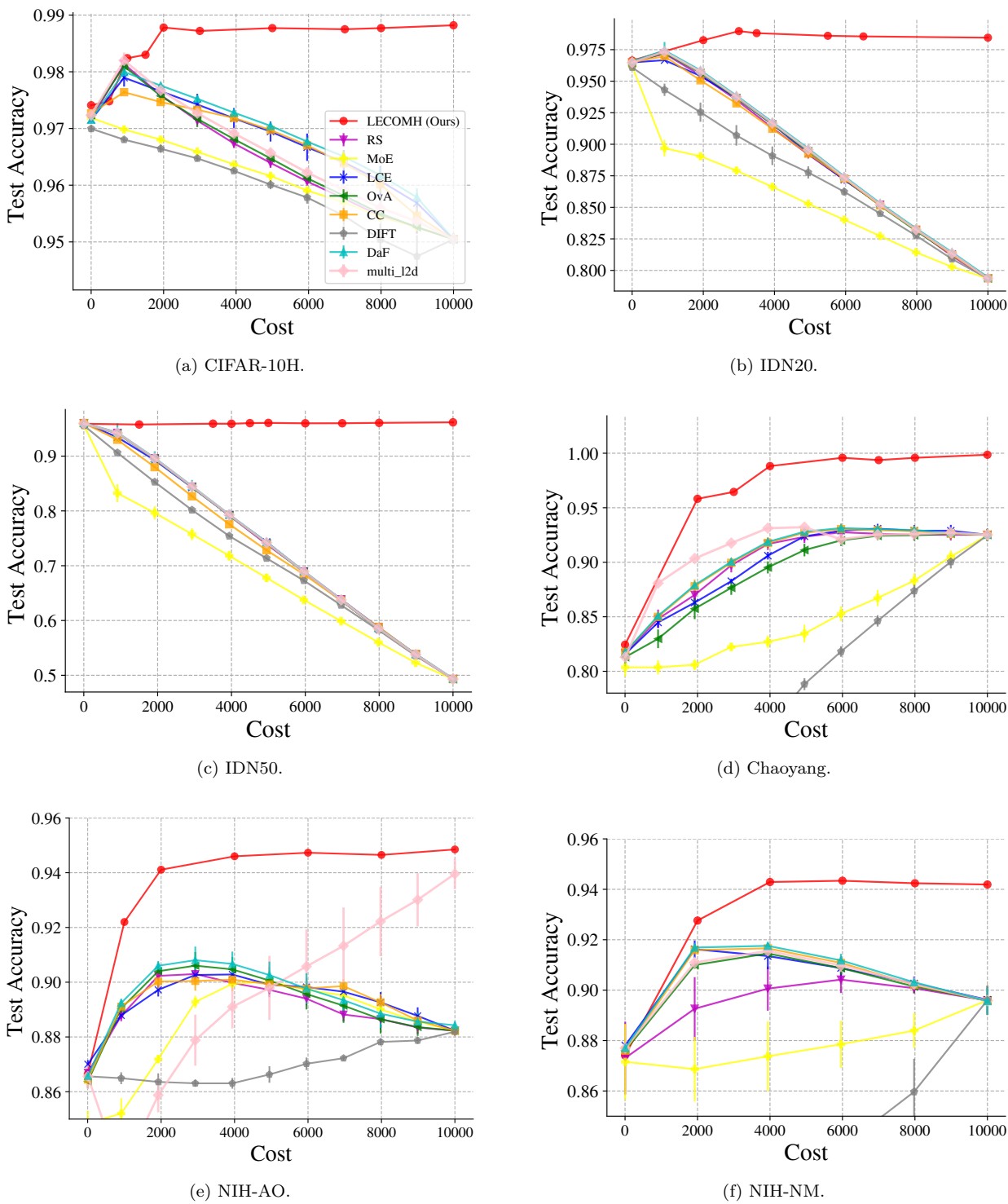

Figure 5: Test accuracy vs. collaboration cost of LECOMH (Ours) and competing SEHAI-CC (Mozannar et al., 2023) and MEHAI-CC (Verma et al., 2023) methods. The SEL2D methods are always pre-trained with LNL techniques, with the single user being simulated with aggregation (majority voting) from the pool of three annotators. Multi_L2D can defer to one of many experts, so we select the label corresponding to the maximum probability of 3 users for each sample to draw the curve. We truncate the accuracy for all methods at cost=10000.

Table 1: Quantitative comparison with the SOTA SEHAI-CC (Mozannar et al., 2023; Charusaie et al., 2024), MEHAI-CC (Hemmer et al., 2022; Verma et al., 2023) and MRL (Goh et al., 2022; Wei et al., 2022; Zhang et al., 2024; Guo et al., 2023; Ibrahim et al., 2023; Cao et al., 2019) methods on the human-AI collabotative classification datasets at 50% coverage. The SEHAI-CC methods are always pre-trained with LNL techniques, with the single user being simulated with random selection from the pool of three annotators. The MEHAI-CC methods are also pre-trained with LNL techniques. The notation * means the results are under zero coverage (i.e., using the annotations from all human experts). CROWDLAB‡ and CROWDLAB denote the results are from combining human experts with pre-trained vanilla ResNet18 prediction and LNL AI model in Sec 3.1.1. The best result per benchmark is marked in bold.

| Methods | Type | CIFAR-10H | IDN20 | IDN50 | IDN{20,30,50} | Chaoyang | NIH-AO | NIH-NM |
|---|---|---|---|---|---|---|---|---|
| AI | LNL | $97.39 \pm 0.16$ | $96.64 \pm 0.04$ | $95.90 \pm 0.25$ | $96.52 \pm 00.11$ | $82.44 \pm 0.20$ | $86.65 \pm 0.35$ | $87.41 \pm 0.19$ |
| Human* | Annotator | 95.10 | 79.36 | 49.30 | 66.22 | 92.47 | 88.23 | 89.60 |
| RS | SEHAI-CC | $96.65 \pm 0.19$ | $89.19 \pm 0.10$ | $73.92 \pm 0.09$ | $96.22 \pm 0.04$ | $92.33 \pm 0.57$ | $89.72 \pm 0.26$ | $90.21 \pm 1.10$ |
| MoE | SEHAI-CC | $96.00 \pm 0.07$ | $85.28 \pm 0.15$ | $67.79 \pm 0.78$ | $95.43 \pm 0.25$ | $83.45 \pm 0.82$ | $90.03 \pm 0.43$ | $87.80 \pm 1.19$ |
| LCE | SEHAI-CC | $96.43 \pm 0.09$ | $89.26 \pm 0.08$ | $74.32 \pm 0.06$ | $96.37 \pm 0.03$ | $92.38 \pm 0.33$ | $89.90 \pm 0.29$ | $90.96 \pm 0.54$ |
| OvA | SEHAI-CC | $97.02 \pm 0.16$ | $89.47 \pm 0.04$ | $74.11 \pm 0.06$ | $96.36 \pm 0.02$ | $91.12 \pm 0.57$ | $90.06 \pm 0.49$ | $91.27 \pm 0.15$ |
| CC | SEHAI-CC | $96.52 \pm 0.10$ | $89.25 \pm 0.13$ | $72.83 \pm 0.11$ | $96.26 \pm 0.00$ | $92.71 \pm 0.47$ | $89.93 \pm 0.31$ | $91.49 \pm 0.33$ |
| DIFT | SEHAI-CC | $96.93 \pm 0.06$ | $87.77 \pm 0.48$ | $71.36 \pm 0.17$ | $95.83 \pm 0.10$ | $78.82 \pm 0.54$ | $86.63 \pm 0.30$ | $83.64 \pm 0.55$ |
| DaF | SEHAI-CC | $97.11 \pm 0.33$ | $88.91 \pm 0.12$ | $75.02 \pm 0.15$ | $95.93 \pm 0.15$ | $92.45 \pm 0.54$ | $90.63 \pm 0.78$ | $91.66 \pm 1.31$ |
| Multi_L2D | MEHAI-CC | $97.22 \pm 0.09$ | $89.54 \pm 0.15$ | $74.05 \pm 0.30$ | $96.47 \pm 0.12$ | $93.84 \pm 0.39$ | $90.67 \pm 0.65$ | $91.17 \pm 0.19$ |
| CET* | MEHAI-CC | $97.76 \pm 0.07$ | $96.13 \pm 0.23$ | $95.18 \pm 0.17$ | $96.77 \pm 0.02$ | $99.20 \pm 0.80$ | $94.14 \pm 0.31$ | $90.57 \pm 0.10$ |
| Majority Vote | MRL | $97.48 \pm 0.00$ | $92.48 \pm 0.05$ | $62.55 \pm 0.44$ | $69.83 \pm 0.00$ | $99.58 \pm 0.00$ | $94.13 \pm 0.00$ | $94.14 \pm 0.00$ |
| Max-MIG | MRL | $84.55 \pm 0.00$ | $85.15 \pm 0.00$ | $84.26 \pm 0.00$ | $84.75 \pm 0.00$ | $70.12 \pm 0.00$ | $77.58 \pm 0.00$ | $77.69 \pm 0.00$ |
| GeoCrowdNet (F) | MRL | $85.71 \pm 0.00$ | $86.62 \pm 0.00$ | $80.11 \pm 0.00$ | $82.32 \pm 0.00$ | $70.32 \pm 0.00$ | $79.35 \pm 0.00$ | $79.66 \pm 0.00$ |
| BayesianIDNT | MRL | $86.64 \pm 0.05$ | $87.03 \pm 0.00$ | $87.31 \pm 0.00$ | $85.62 \pm 0.40$ | $72.09 \pm 0.00$ | $80.22 \pm 0.00$ | $81.34 \pm 0.00$ |
| CCC | MRL | $85.52 \pm 0.05$ | $75.17 \pm 0.00$ | $44.64 \pm 0.00$ | $56.16 \pm 0.00$ | $72.33 \pm 0.00$ | $79.61 \pm 0.00$ | $80.55 \pm 0.00$ |
| UnionNet-B | MRL | $93.34 \pm 0.45$ | $95.59 \pm 0.11$ | $93.54 \pm 0.21$ | $94.11 \pm 0.00$ | $74.08 \pm 0.92$ | $85.94 \pm 0.58$ | $87.50 \pm 0.52$ |
| CROWDLAB‡ | MRL | $97.96 \pm 0.00$ | $92.35 \pm 0.00$ | $60.96 \pm 0.00$ | $71.22 \pm 0.00$ | $99.58 \pm 0.00$ | $94.05 \pm 0.00$ | $94.05 \pm 0.00$ |
| CROWDLAB | MRL | $97.72 \pm 0.02$ | $92.40 \pm 0.18$ | $61.95 \pm 0.31$ | $79.00 \pm 0.00$ | $99.58 \pm 0.00$ | $94.09 \pm 0.00$ | $92.61 \pm 0.00$ |
| LECOMH | MEHAI-CC | $\mathbf{98.77 \pm 0.10}$ | $\mathbf{98.82 \pm 0.15}$ | $\mathbf{96.05 \pm 0.05}$ | $\mathbf{97.15 \pm 0.04}$ | $\mathbf{99.58 \pm 0.42}$ | $\mathbf{94.63 \pm 0.12}$ | $\mathbf{94.19 \pm 0.05}$ |

where coverage can be defined as the fraction of the test samples predicted by the AI model. To calculate the SEHAI-CC's collaboration coverage, we follow the procedure presented in (Mozannar et al., 2023) by sorting the testing images based on their rejection scores and then adjusting the threshold of the quantile used for annotating these testing cases by users. We also compare LECOMH with methods that defer to multiple experts (MEHAI-CC), including classifier and expert team (CET) (Hemmer et al., 2022), and learning to defer to multiple experts (Multi_L2D) (Verma et al., 2023) in our setting. For a fair comparison, all classification backbones for the {SE,ME}HAI-CC methods have the same architecture. All {SE,ME}HAI-CC methods rely on LNL pre-training because they provide better results for all cases. For all {SE,ME}HAI-CC methods, hyper-parameters are set as previously reported in (Mozannar et al., 2023; Hemmer et al., 2022; Verma et al., 2023). We also provide the results of SOTA LNL approaches, such as ProMix (Wang et al., 2023) (CIFAR-10N), InstanceGM (Garg et al., 2023) (CIFAR10-IDN), NSHE (Zhu et al., 2021a) (Chaoyang) and NVUM (Liu et al., 2022) (NIH datasets), and SOTA MRL approaches (Wei et al., 2022; Zhang et al., 2024; Guo et al., 2023; Ibrahim et al., 2023; Cao et al., 2019), such as majority voting, CCC (Zhang et al., 2024), BayesianIDNT (Guo et al., 2023), GeoCrowdNet (F) (Ibrahim et al., 2023), Max-MIG (Cao et al., 2019), UnionNet-B (Wei et al., 2022) and CROWDLAB (Goh et al., 2022). Note that when considering CROWDLAB as an isolated baseline, its AI model is the pre-trained vanilla ResNet18 trained with early stopping, not the LNL models mentioned in Section 5.1.

## 5.4 Results

Fig. 4 showcases the accuracy of LECOMH and baseline methods as a function of coverage in all benchmarks. In these graphs, the full coverage point (i.e., coverage equals to 1, meaning only the AI model makes prediction) represents the performance of the LNL pre-trained methods, possibly fine-tuned by the HAI-CC method (resulting in slight variations from the original LNL accuracy). Moreover, the results with minimum coverage (i.e., coverage equals to 0) represent the performance of experts alone.

Table 2: Quantitative comparison of the best accuracy of the SOTA SEHAI-CC (Mozannar et al., 2023; Charusaie et al., 2024), MEHAI-CC (Hemmer et al., 2022; Verma et al., 2023) and MRL (Goh et al., 2022; Wei et al., 2022; Zhang et al., 2024; Guo et al., 2023; Ibrahim et al., 2023; Cao et al., 2019) methods on the human-AI collabotative classification datasets. The SEHAI-CC methods are always pre-trained with LNL techniques, with the single user being simulated with random selection from the pool of three annotators. The MEHAI-CC methods are also pre-trained with LNL techniques. The notation * means the results are under zero coverage (i.e., using the annotations from all human experts). CROWDLAB‡ and CROWDLAB denote the results are from combining human experts with pre-trained vanilla ResNet18 prediction and LNL AI model in Sec 3.1.1. The best result per benchmark is marked in bold.

| Methods | Type | CIFAR-10H | IDN20 | IDN50 | IDN{20,30,50} | Chaoyang | NIH-AO | NIH-NM |
|---|---|---|---|---|---|---|---|---|
| AI | LNL | 97.39 ± 0.16 | 96.64 ± 0.04 | 95.90 ± 0.25 | 96.52 ± 00.11 | 82.44 ± 0.20 | 86.65 ± 0.35 | 87.41 ± 0.19 |
| Human | Annotator | 95.10 | 79.36 | 49.30 | 66.22 | 92.47 | 88.23 | 89.60 |
| RS | SEHAI-CC | 98.14 ± 0.10 | 97.18 ± 0.01 | 95.99 ± 0.04 | 96.22 ± 0.04 | 92.75 ± 0.51 | 90.30 ± 0.21 | 90.42 ± 0.25 |
| MoE | SEHAI-CC | 97.19 ± 0.03 | 96.18 ± 0.12 | 95.64 ± 0.04 | 95.43 ± 0.25 | 92.47 ± 0.54 | 90.28 ± 0.23 | 89.60 ± 0.59 |
| LCE | SEHAI-CC | 97.91 ± 0.13 | 96.67 ± 0.01 | 96.06 ± 0.02 | 96.37 ± 0.03 | 93.08 ± 0.44 | 90.60 ± 0.50 | 91.62 ± 0.35 |
| OvA | SEHAI-CC | 98.11 ± 0.11 | 97.24 ± 0.03 | 95.95 ± 0.02 | 96.36 ± 0.02 | 92.61 ± 0.52 | 90.10 ± 0.43 | 91.51 ± 0.06 |
| CC | SEHAI-CC | 97.58 ± 0.08 | 96.98 ± 0.12 | 96.02 ± 0.02 | 96.26 ± 0.00 | 93.03 ± 0.53 | 88.19 ± 0.16 | 91.85 ± 0.04 |
| DIFT | SEHAI-CC | 96.96 ± 0.07 | 96.09 ± 0.21 | 95.58 ± 0.14 | 95.83 ± 0.10 | 92.47 ± 0.54 | 90.03 ± 0.43 | 89.60 ± 0.59 |
| DaF | SEHAI-CC | 98.22 ± 0.42 | 97.55 ± 1.21 | 96.02 ± 0.01 | 95.93 ± 0.25 | 93.45 ± 0.48 | 90.63 ± 0.78 | 91.66 ± 1.31 |
| Multi_L2D | MEHAI-CC | 98.17 ± 0.07 | 97.32 ± 0.06 | 95.99 ± 0.04 | 96.47 ± 0.12 | 98.14 ± 0.05 | 91.80 ± 1.05 | 91.55 ± 0.19 |
| CET* | MEHAI-CC | 97.76 ± 0.07 | 96.13 ± 0.23 | 95.18 ± 0.17 | 96.77 ± 0.02 | 99.20 ± 0.80 | 94.14 ± 0.31 | 90.57 ± 0.10 |
| Majority Vote | MRL | 97.48 ± 0.00 | 92.48 ± 0.05 | 62.55 ± 0.44 | 69.83 ± 0.00 | 99.58 ± 0.00 | 94.13 ± 0.00 | 94.14 ± 0.00 |
| Max-MIG | MRL | 84.55 ± 0.00 | 85.15 ± 0.00 | 84.26 ± 0.00 | 84.75 ± 0.00 | 70.12 ± 0.00 | 77.58 ± 0.00 | 77.69 ± 0.00 |
| GeoCrowdNet (F) | MRL | 85.71 ± 0.00 | 86.62 ± 0.00 | 80.11 ± 0.00 | 82.32 ± 0.00 | 70.32 ± 0.00 | 79.35 ± 0.00 | 79.66 ± 0.00 |
| BayesianIDNT | MRL | 86.64 ± 0.05 | 87.03 ± 0.00 | 87.31 ± 0.00 | 85.62 ± 0.40 | 72.09 ± 0.00 | 80.22 ± 0.00 | 81.34 ± 0.00 |
| CCC | MRL | 85.52 ± 0.05 | 75.17 ± 0.00 | 44.64 ± 0.00 | 56.16 ± 0.00 | 72.33 ± 0.00 | 79.61 ± 0.00 | 80.55 ± 0.00 |
| UnionNet-B | MRL | 93.34 ± 0.45 | 95.59 ± 0.11 | 93.54 ± 0.21 | 94.11 ± 0.00 | 74.08 ± 0.92 | 85.94 ± 0.58 | 87.50 ± 0.52 |
| CROWDLAB‡ | MRL | 97.96 ± 0.05 | 92.35 ± 0.00 | 60.96 ± 0.00 | 71.22 ± 0.00 | 99.58 ± 0.00 | 94.05 ± 0.00 | 94.05 ± 0.00 |
| CROWDLAB | MRL | 97.72 ± 0.02 | 92.40 ± 0.18 | 61.95 ± 0.31 | 79.00 ± 0.00 | 99.58 ± 0.00 | 94.09 ± 0.00 | 92.61 ± 0.00 |
| LECOMH | MEHAI-CC | **98.82 ± 0.05** | **98.87 ± 0.10** | **96.08 ± 0.02** | **97.31 ± 0.14** | **99.79 ± 0.21** | **94.85 ± 0.15** | **94.30 ± 0.09** |

Despite using the same backbone models trained with leading LNL methods and obtaining consensus labels from CROWDLAB, the observed differences highlight the effectiveness of our proposed LECOMH in human-AI collaborative classification techniques in comparison with all baseline methods. More specifically, the results in Fig. 4 highlight LECOMH's ability not only to minimise the impact of expert prediction errors, but also to integrate expert information.

In fact, compared to all baselines, LECOMH is the only method that consistently achieves high human-AI collaborative classification accuracy across different levels of coverage, surpassing human and AI accuracies in all benchmarks.

A more fine-grained analysis of LECOMH's results in Fig. 4 suggest that: 1) for scenarios where human experts have relatively higher or similar accuracy as LNL methods (e.g., CIFAR-10H, Chaoyang, NIH-Aispace opacity and NIH-Nodule/mass), HAI-CC methods can offer significant gains; but 2) for scenarios where users have low accuracy (e.g., IDN20 and IDN50) and LNL methods provide highly accurate predictions, HAI-CC methods offer little improvements over the LNL classifier results. Hence, as already concluded in (Rastogi et al., 2023), it is important to study the conditions that enable HAI-CC to thrive. Nevertheless, it is interesting to note that for LECOMH, if we set coverage at 50%, we are always guaranteed to obtain the best possible HAI-CC performance that is usually better (or at least comparable, as in IDN50) than AI or user alone.

In addition to the accuracy versus coverage graphs of Fig. 4, we also provide quantitative analyses in Tables 1 and 2. In particular, the results in Table 1 are the classification accuracy at a fixed coverage=50%, while the ones in Table 2 show the best possible accuracy. The notation * denotes the results of those methods are under the condition of zero coverage, which means that the prediction depends on the manual annotations from all human experts. Note that LECOMH surpasses all SE- and ME-HAI-CC methods. LECOMH also outperforms MRL, AI, and human annotators without collaborating with all of the human experts, especially in NIH-AO and NIH-NM datasets. Notably, in CIFAR-10H, IDN20 and IDN50 benchmarks, the LNL AI

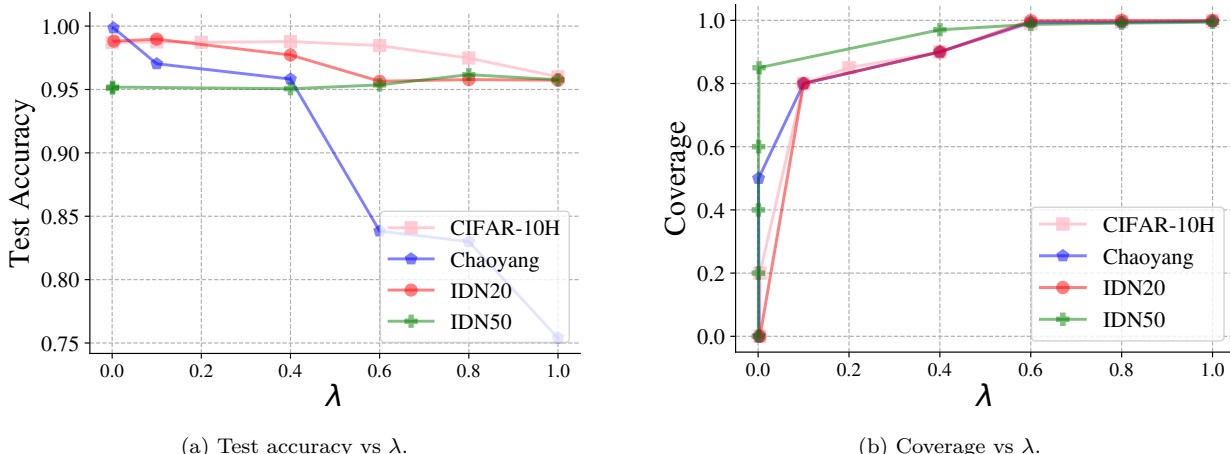

(a) Test accuracy vs $\lambda$.  (b) Coverage vs $\lambda$.

Figure 6: Test accuracy vs coverage as a function of $\lambda$ in Eq. (4) that weights the collaboration cost in our optimisation.

model has higher accuracy than humans, while in Chaoyang, NIH-AO and NIH-NM datasets, the LNL AI model has lower accuracy than pathologists.

Tables 3 and 4 present human-AI classification results at the coverage rates of approximately 50% on the test set of Chaoyang and CIFAR-10H, respectively. Cases include the test image, human-provided labels ($\mathcal{M}$), LNL AI model prediction ($f_\theta(.)$), prediction probability vector by the Human-AI Selection Module ($g_\phi(.)$), final prediction by the Collaboration Module ($h_\psi(.)$), and ground truth (GT) label. Notably, when the AI model or the users make individual mistakes, the final prediction tends to be correct, highlighting system robustness. $g_\phi(.)$ often assigns high probability to the AI model when the AI model appears to be correct, indicating reliance on AI predictions. On the other hand, when the AI model seems to be incorrect, then $g_\phi(.)$ often assigns low probability to the AI model, and high probabilities to the human-AI collaborative classifications, suggesting a reliance on users.

Additionally, to better demonstrate LECOMH's effectiveness, we further evaluated the cost differences caused by assigning one or multiple experts to collaborate on each sample. While accuracy-coverage curves have been widely used to evaluate model performance, at the same coverage level, the cost varies depending on whether one or multiple experts are involved in the collaboration for each sample. Therefore, we present the accuracy-cost curve in Figure 5, which more intuitively reflects classification accuracy at different cost levels, providing a more comprehensive evaluation of the model's performance. We define a system cost, where a cost of 1 denotes that a single expert provides a prediction in the decision process. For CIFAR-10, the total cost computation is governed by Eq. (5), with the parameter $\lambda$ in Eq. (4) adjusted during the training phase to influence LECOMH's cost considerations. This results in a minimum cost of 0 (all testing cases predicted by AI alone) and a maximum cost of 30000 (all testing cases predicted by AI + 3 users or deferred to 3 users) for 10K test images. For the Chaoyang and NIH datasets, the cost scale is normalized from the original span to a broader range of $[0, 3 \times 10^4]$ to enable easier comparisons across different datasets. Single-expert methods have a total cost in $[0, 10^4]$ as only one user per image is allowed. LECOMH assesses accuracy within the cost range of $[0, 10^4]$, and for multi-expert methods, the accuracy plot is truncated at cost = 10000 for consistency in comparative analysis when costs exceed this value. LECOMH demonstrates higher classification accuracy than all competing HAI-CC methods across all collaboration costs and benchmarks. For most competing methods, the accuracy at cost = 0 primarily reflects the performance of the LNL pre-trained model. Accuracy generally increases as collaboration cost rises, peaks at some cost $< 10,000$, and then gradually decreases until it matches the accuracy of human annotators at cost = 10000. Notably, when the AI model outperforms humans (e.g., CIFAR-10H), both LECOMH and competing methods achieve more accurate predictions than humans or AI alone. In contrast, when human accuracy surpasses AI (e.g., Chaoyang), SEHAI-CC methods are constrained by expert information, resulting in limited accuracy improvements. However, MEHAI-CC

Table 3: Human-AI classification (coverage $\approx 50\%$) of Chaoyang test samples, where $\mathcal{M}$ denotes human labels, $f_\theta(.)$ is the LNL AI model's classification, $g_\phi(.)$ represents the Human-AI Selection prediction probability vector for [AI prediction (1st value), AI + 1 User (2nd value), AI + 2 Users (3rd value), AI + 3 Users (4th value)], $h_\psi(.)$ is the final prediction from the Collaboration Module, and GT denotes the ground truth label.

| Image | $\mathcal{M}$ | $f_\theta(.)$ | $g_\phi(.)$ | $h_\psi(.)$ | GT |
|---|---|---|---|---|---|
| | adenocarcinoma, normal, adenocarcinoma | normal | [0.48, 0.04, 0.03, 0.45] | adenocarcinoma | adenocarcinoma |
| | normal, normal, normal | adenocarcinoma | [0.45, 0.06, 0.07, 0.42] | normal | normal |
| | adenoma, adenoma, adenoma | adenocarcinoma | [0.49, 0.07, 0.12, 0.33] | adenoma | adenoma |
| | serrated, normal, serrated | serrated | [0.82, 0.04, 0.03, 0.10] | serrated | serrated |
| | adenoma, adenoma, serrated | adenoma | [0.82, 0.01, 0.02, 0.14] | adenoma | adenoma |
| | normal, adenocarcinoma, adenocarcinoma | adenocarcinoma | [0.87, 0.03, 0.03, 0.07] | adenocarcinoma | adenocarcinoma |

methods generally outperform SEHAI-CC approaches as collaboration costs increase, though they fall short of LECOMH's accuracy at lower cost levels.

## 5.5 Ablation Studies

### 5.5.1 Study of each LECOMH component

In this subsection, we analyse the effect of the following LECOMH components: 1) LNL methods to pre-train the the AI model, 2) multi-rater learning, 3) the role of multiple users for the collaboration, and 4) the integration of the learning to complement module. Table 5 shows the results of different settings where in each setting, one of the factors is replaced by a baseline approach.

**LNL pre-training** By replacing the LNL pre-training by a regular classifier pre-training with early stopping using the noisy labels, we form the LECOMH‡. Note in Table 5 that this model performs poorly in scenarios where the LNL AI model surpasses original human labels, such as CIFAR-10H, IDN20 and IDN50. The absence of LNL pre-training is particularly more severe at high label noise rates (e.g., IDN50 with 50% noise rate).

**Multi-rater learning** The replacement of MRL approaches, such as CROWDLAB, by simpler methods to obtain consensus labels (e.g., majority voting or sampling a random label as consensus label) forms the approach denoted as LECOMH⋄. As shown in Table 5, the absence of MRL negatively impacts LECOMH across various cases, with a more pronounced effect at high label noise rates (e.g., IDN50) and evident even in other lower noise rate scenarios when using a random label as consensus (e.g. Chaoyang). To further emphasise the value of MRL, we show the accuracy of the consensus label in the training set produced by majority vote, UnionNet (Wei et al., 2022) and CROWDLAB (Goh et al., 2022) in Table 6, where results confirm that the MRL approaches, and in particular CROWDLAB, produce more accurate consensus label.

Table 4: Human-AI classification (coverage $\approx 50\%$) of CIFAR-10H test samples, where $\mathcal{M}$ denotes human labels, $f_\theta(.)$ is the LNL AI model's classification, $g_\phi(.)$ represents the Human-AI Selection prediction probability vector for [AI prediction (1st value), AI + 1 User (2nd value), AI + 2 Users (3rd value), AI + 3 Users (4th value)], $h_\psi(.)$ is the final prediction from the Collaboration Module, and GT denotes the ground truth label.

| Image | $\mathcal{M}$ | $f_\theta(.)$ | $g_\phi(.)$ | $h_\psi(.)$ | **GT** |
|---|---|---|---|---|---|
| | cat, cat, cat | dog | [0.57, 0.05, 0.07, 0.31] | cat | cat |
| | bird, bird, bird | frog | [0.57, 0.05, 0.04, 0.34] | bird | bird |
| | truck, car, plane | plane | [0.70, 0.06, 0.04, 0.20] | plane | plane |
| | dog, horse, frog | frog | [0.71, 0.06, 0.04, 0.19] | frog | frog |
| | truck, truck, car | car | [0.79, 0.03, 0.05, 0.13] | car | car |
| | ship, deer, ship | deer | [0.87, 0.02, 0.02, 0.09] | deer | deer |

Table 5: Accuracy results for the ablation experiments at 50% coverage (the best accuracy is shown inside brackets). LNL, MRL, ME, L2C denote the utilisation of LNL pre-trained model, the introduction of consensus label via CROWDLAB, the integration of multiple experts, and the cooperation of human experts and AI prediction, respectively. The last row shows the final LECOMH's results.

| Methods | LNL | MRL | ME | L2C | CIFAR-10H | IDN20 | IDN50 | Chaoyang | NIH-AO | NIH-NM |
|---|---|---|---|---|---|---|---|---|---|---|
| Human* | | | | | 95.10 | 79.36 | 49.30 | 92.47 | 88.23 | 89.60 |
| LNL | √ | ✗ | ✗ | ✗ | 97.41 | 96.64 | 95.90 | 82.44 | 86.65 | 87.41 |
| LECOMH‡ | ✗ | √ | √ | √ | 97.71 (97.95) | 96.94 (96.94) | 67.39 (95.90) | 97.38 (99.58) | 93.56 (94.65) | 93.67 (94.19) |
| LECOMH◊ | √ | ✗ | √ | √ | 98.22 (98.25) | 97.69 (98.01) | 80.52 (95.90) | 96.83 (100.00) | 92.78 (94.44) | 92.27 (94.12) |
| LECOMH† | √ | √ | ✗ | √ | 97.83 (97.83) | 97.71 (97.80) | 95.79 (95.90) | 92.96 (94.15) | 91.65 (92.61) | 92.46 (93.27) |
| LECOMH§ | √ | √ | √ | ✗ | 86.01 (98.17) | 85.93 (93.37) | 54.63 (69.05) | 69.34 (99.58) | 85.93 (93.37) | 85.32 (93.17) |
| LECOMH | √ | √ | √ | √ | **98.77 (98.87)** | **98.82 (98.97)** | **96.05 (96.10)** | **99.58 (100.00)** | **94.63 (95.00)** | **94.19 (94.39)** |

**Multiple users for the collaboration** By replacing the collaboration with multiple users with a collaboration with a single user, we form LECOMH†. This new model highlights the crucial role of multiple users, particularly in low-noise rate scenarios (e.g., CIFAR-10H, Chaoyang, IDN20, NIH-AO and NIH-NM). In these cases, LECOMH with three users consistently outperforms LECOMH with a single user. However, for IDN50, the reliance on multiple users does not significantly affect LECOMH, indicating that, for high noise rates, the choice between the collaboration with single or multiple users does not have a significant impact.

**Integration of the learning to complement module** By replacing the collaboration module by a simpler majority voting module, we build the model LECOMH§. As shown in Table 5, the performance of LECOMH§ is reduced significantly compared to the ones with the collaboration module, highlighting the importance of the collaboration module when designing an human-AI system.

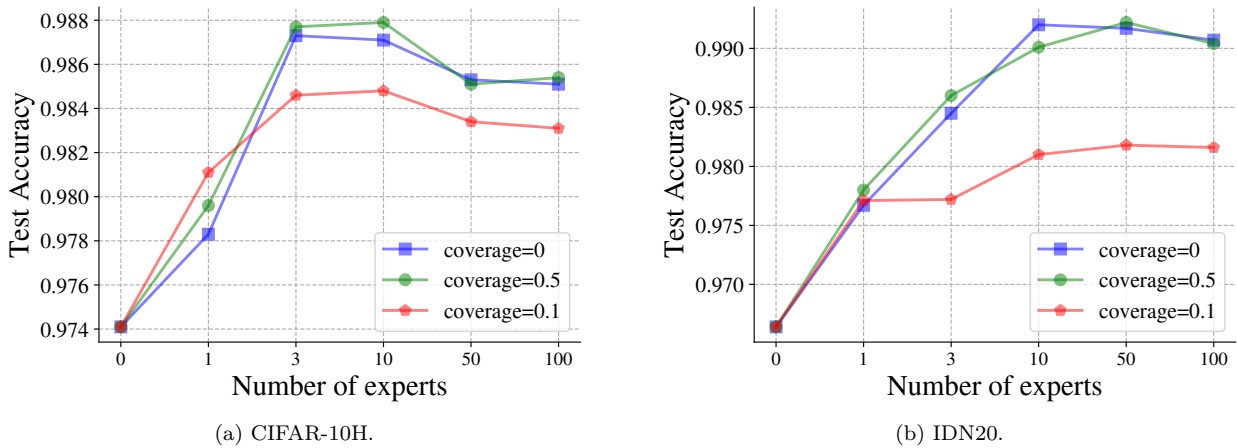

(a) CIFAR-10H.

(b) IDN20.

Figure 7: Test accuracy vs number of experts at different coverage.

Table 6: Accuracy of the consensus label from majority vote, UnionNet (Wei et al., 2022) and CROWD-LAB (Goh et al., 2022) in the training set.

|  | CIFAR-10N | IDN20 | IDN50 | Chaoyang | NIH-AO | NIH-NM |
|---|---|---|---|---|---|---|
| Majority Vote | 0.91 | 0.94 | 0.69 | 0.99 | 0.94 | 0.94 |
| UnionNet | 0.92 | 0.94 | 0.90 | 0.99 | 0.86 | 0.87 |
| CROWDLAB | 0.98 | 0.99 | 0.98 | 0.99 | 0.94 | 0.96 |

### 5.5.2 Cost, Scalability and Training Time

**Varying the weight of cost hyperparameter $\lambda$ in Eq. (4)** To study the contribution of the cost hyperparameter to the performance of LECOMH, we run several experiments using different values for $\lambda$ and measuring the performance on several benchmarks. The results in Fig. 6 show the tradeoff between coverage and accuracy when varying $\lambda$. Smaller values for $\lambda$ imply higher accuracy results, as shown in Fig. 6a, but it also leads to smaller coverage (see Fig. 6b), which is expected due to the lower cost from the querying of human experts. On the other hand, higher values for $\lambda$ show lower accuracy and larger coverage.

**Scalability with many users** In Fig. 7, we explore the scalability of LECOMH to a large number of users. Instead of collaborating with a maximum of three users, as demonstrated in previous experiments, we show in Fig. 7 the performance of LECOMH, on CIFAR-10H and IDN20, when it collaborates with between 0 and 100 users. The training with CIFAR-10N required simulating numerous users beyond the three available, which is achieved by learning user-specific label-transition matrices and synthesising labels from these matrices. Notably, CIFAR-10H's performance is unsurprisingly optimised for three users, reflecting the synthetic nature of redundant users. On the other hand, for IDN20, accuracy peaks at around 10 to 50 users, suggesting a correlation between problem difficulty and the required number of users for effective collaboration. Also note that the run-time complexity of our optimisation has a linear increase in terms of the number of users. In practice, the training time increases from 29s (3 users) to 30s (100 users) per epoch for both CIFAR-10H and IDN20.

**Scalability with different expert reliability** An important real-world scenario occurs when users exhibit varying levels of accuracy, departing from the assumption of a homogeneous noise rate used in previous experiments. This variation is evident in the Chaoyang and NIH datasets, as shown in Figs. 4 and 5. To further evaluate the effectiveness of our method in this challenging setting, we conducted an additional experiment, illustrated in Tables 1 and 2 and Fig. 8, where varying levels of expert accuracy were simulated with IDN noise rates of 20%, 30%, and 50%. The results demonstrate that under conditions of high coverage and low cost, LECOMH achieves superior performance compared to competing methods. However, as coverage

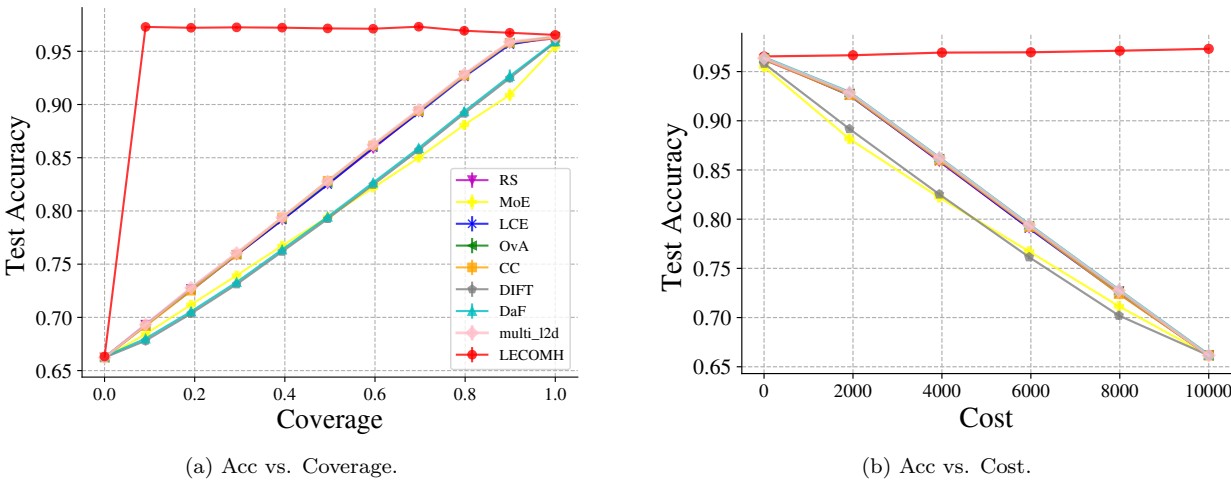

(a) Acc vs. Coverage.

(b) Acc vs. Cost.

Figure 8: Collaboration with three experts showing different levels of noise with 20%, 30%, and 50% IDN. The graphs show test accuracy vs. coverage (a) and collaboration cost (b) of LECOMH (Ours) and competing SEHAI-CC (Mozannar et al., 2023) and MEHAI-CC (Verma et al., 2023) methods on IDN{20,30,50} dataset.

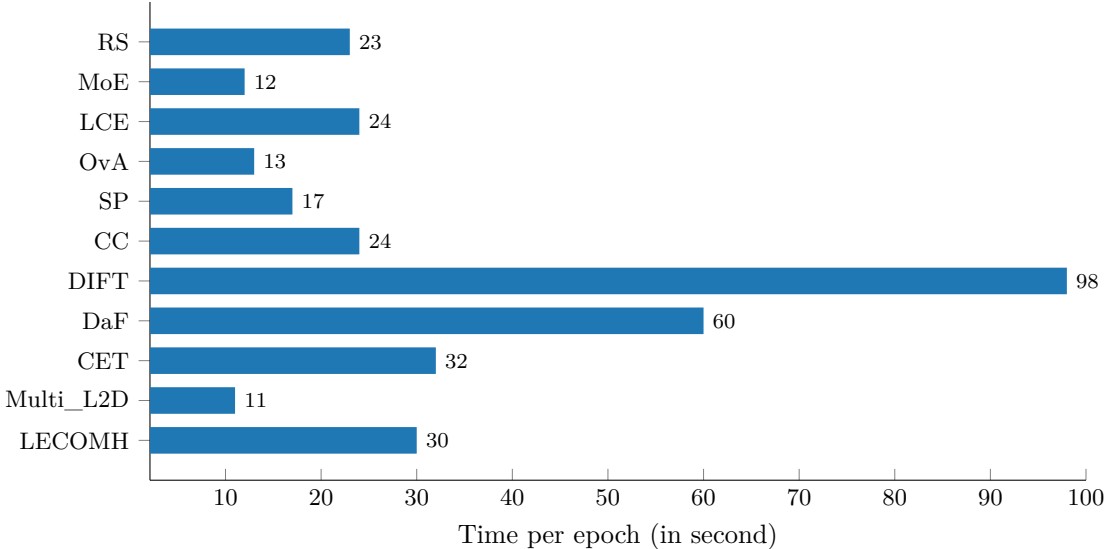

Figure 9: Training time/epoch of LECOMH and competing methods on CIFAR-10N.

decreases and cost increases, the performance of competing methods significantly deteriorates due to the lower average accuracy of the human experts relative to the AI classifier. In contrast, LECOMH effectively collaborates with the users even in these scenarios, though it does so with reduced confidence levels, ensuring more balanced performance across varying conditions.

**Training time** In Fig. 9, we compare the training time per epoch of LECOMH and SOTA HAI-CC approaches on CIFAR-10N. Notice that our approach has similar training time compared to other learning to complement approaches, but it has slightly larger training time in comparison to the simpler learning to defer methods.

# 6 Conclusion

We introduced the first human-AI collaborative classification method that can be trained exclusively from multiple noisy labels to maximise the collaborative classification accuracy of teams of AI and multiple experts, while minimising the collaboration costs, measured by the number of human experts used in HAI-CC. Additionally, we introduce two new HAI-CC benchmarks that rely on multiple noisy labels in the training and testing sets. Comparative analysis with SOTA HAI-CC methods on our benchmarks demonstrates that LECOMH consistently outperforms the competition, showcasing increased accuracy at comparable collaboration costs. Importantly, LECOMH stands out as the only method enhancing expert labellers and isolated noisy-label learning methods across all benchmarks.

The major limitation of LECOMH is that it is unbiased to any labeller, which implicitly assumes that labellers have similar performance. Even though this limitation mitigates the complexity involved in the combinatorial selection of specific subsets of labellers, we plan to address this issue by exploring a strategy where labellers can be characterised and selected during training and testing, so the system will be able to better adapt to the user's performance. A positive side effect of this issue is that LECOMH does not rely on specific user IDs, which helps mitigate privacy concerns related to expert collaboration with the system. Another limitation of LECOMH is its dependence on a sampling-based method in the testing phase for selecting the number of users to collaborate with the AI classifier. Such stochastic selection is needed because the arg max operator on the module's output almost always favours selecting the AI model alone for predictions, as shown in Tables 3 and 4. On the other hand, by using Gumbel-Softmax sampling from the categorical distribution of the module's output, we can achieve a more balanced and accurate classification. We plan to address these issues by selecting the specific experts to collaborate with the AI model in both training and testing phases and having a more reliable coverage constraint for training to enable a deterministic selection of the users to collaborate with the AI classifier.

One excessive simplification of our method is that it assumes the cost of a user input to be a flat value of "1", representing the initial step in formulating HAI-CC methods. However, more sophisticated cost models need to be developed to account for the impacts of false positives and false negatives, particularly in the context of patient-specific conditions (e.g., the high cost of a false negative). We are actively collaborating with health economists to study these factors and plan to incorporate such nuanced cost models in future iterations of this work.

By improving the performance of users who interact with AI systems, we believe that LECOMH has a potential benefit to society given the more accurate outcomes produced by the system and the generally improved performance of labellers. Nevertheless, LECOMH may also have potential negative societal impacts. For example, users may become overconfident on the AI model, which can de-skill some professionals (e.g., radiologists or pathologists). Our future work will integrate techniques to prevent such de-skilling process from happening.

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

# A Appendix

## A.1 Experiments on Other Real-world Dataset

**MiceBone** (Schmarje et al., 2019; 2022a;b) has 7,240 second-harmonic generation microscopy images, with each image being annotated by one to five professional annotators, where the annotation consists of one of three possible classes: *similar collagen fiber orientation, dissimilar collagen fiber orientation, and not of interest due to noise or background.* Only 8 out of 79 annotators label the whole dataset. We, therefore, use these 8 annotators to represent the experts in our experiment. Using the majority vote as the ground truth, the accuracy of those 8 experts are from 84% to 86%. As the dataset is divided into 5 folds, we use the first 4 folds as the training set, and the remaining fold as the test set.

To further assess the effectiveness of our method in real-world scenarios, we conducted an additional experiment using the Micebone dataset, as shown in Fig. 10. The results demonstrate that under conditions of large number of specific experts, LECOMH achieves superior performance compared to competing methods. Moreover, since the accuracy gap between these experts is relatively small, performance gains from collaborating with more experts become marginal.

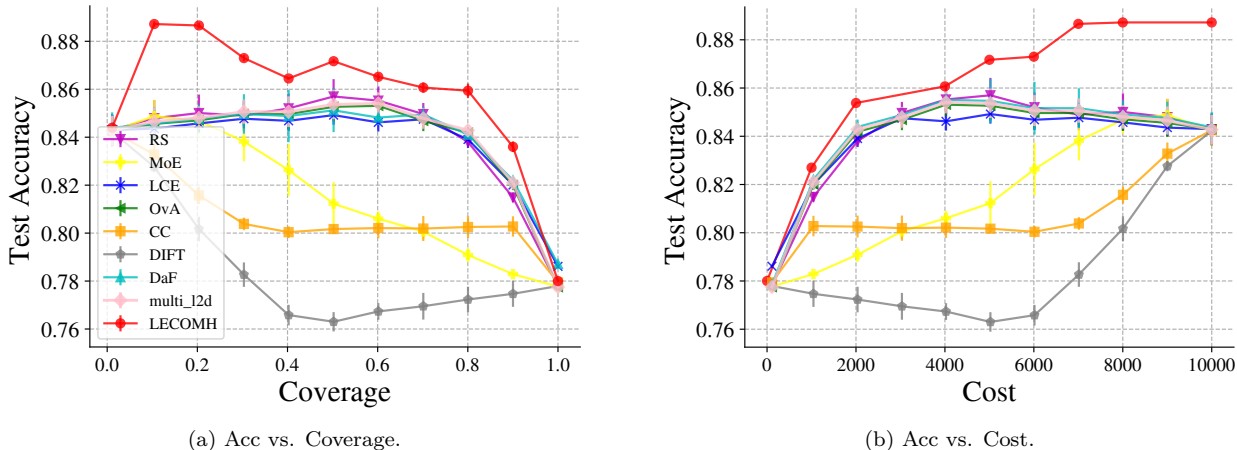

(a) Acc vs. Coverage.

(b) Acc vs. Cost.

Figure 10: The graphs show test accuracy vs. coverage (a) and collaboration cost (b) of LECOMH (Ours) and competing SEHAI-CC (Mozannar et al., 2023) and MEHAI-CC (Verma et al., 2023) methods on Micebone dataset.

