# OpenReview forum: "Learning to Complement with Multiple Humans"
_TMLR — Rejected by TMLR_

### Review · Reviewer_gDA4 · 2024-12-15

**Summary Of Contributions:**

This paper addresses the problem of training and testing with multiple, error-prone human annotators. Instead of relying solely on these annotators to build a training dataset for a neural network, their inputs can also be leveraged during testing. To achieve this, the authors introduce the *Learning to Complement with Multiple Humans* (LECOMH) approach, which integrates techniques from *learning with noisy labels*(LNL) and *multi-rater learning* (MRL) into one pipeline, whose training process consists of two stages:

- **Stage 1:** A neural network is trained using an LNL technique and randomly sampled noisy labels from the training set. The resulting LNL model then predicts labels for each training instance. These predictions are combined with the annotators’ noisy labels using CROWDLAB, a multi-rater learning technique, to produce consensus labels.
- **Stage 2:** Using the consensus labels as proxies for ground truth, LECOMH jointly trains two additional models: A human-AI selection model that decides whether the LNL model’s prediction alone is sufficient or if additional human annotations are needed. A collaboration model that combines the LNL prediction and, if requested, human annotations to approximate the consensus labels. Both models are jointly trained with a cross-entropy loss, and a regularization term is added to discourage excessive reliance on human annotations.

An empirical evaluation on multiple real-world datasets demonstrates that LECOMH outperforms conventional human-AI collaboration, LNL, and MRL approaches.

**Audience:**

Yes

**Broader Impact Concerns:**

The authors sufficiently point out potential implications of the approach as part of their conclusion.

**Claims And Evidence:**

No

**Requested Changes:**

I would appreciate it if the authors could address the aforementioned weakness and clarify the questions raised. In particular, I suggest training stronger baselines using LNL with all available labels, and providing a more detailed discussion on how to select the $\lambda$ parameter in practice when adhering to a specific collaboration cost budget. If I have misunderstood any aspect, I am willing to reconsider my perspective on these issues. Looking forward to a constructive discussion!

**Strengths And Weaknesses:**

Below, I list the strengths and weaknesses according to their importance, starting with the most important one and ending with the least important one.

---

**Strengths:**
- The proposed LECOMH pipeline addresses a relevant yet underexplored research topic by not only training with class labels provided by multiple, error-prone human annotators, but also by leveraging an intelligent and cost-efficient collaboration with these annotators during the testing phase.

- Empirical evaluations on numerous well-known real-world datasets—where class labels are noisy and human-provided—demonstrate LECOMH’s superior performance compared to several related approaches.

- An ablation study confirms the value of each individual component of LECOMH, thereby validating the authors’ design choices.

- The paper is clearly written, well-structured, and supported by explanatory illustrations, making it easily understandable for the reader.

**Weaknesses:**
- In Section 3.1.1, the authors state:

  > This LNL training uses the training set, where the noisy label of image $x_i$ is randomly selected as one of the experts’ annotations in $\mathcal{M}_i$.

  The rationale for this approach is not immediately clear. Why rely on a single randomly chosen annotation for training instead of leveraging
  all available labels? For instance, one might aggregate multiple annotations to produce a consensus label, potentially improving the
  robustness and accuracy of the LNL model.

  A similar concern arises from the statement in Section 5.1.3:

  > Note that when considering CROWDLAB as an isolated baseline, its AI model is the pre-trained vanilla ResNet18 trained with early
  stopping, not the LNL models mentioned in Section 5.1.1.

  Here, it would be more realistic and informative to compare against baselines that incorporate all available class labels and the best possible
  LNL techniques. Such baselines could be combined with MRL methods like CROWDLAB or majority voting to yield stronger reference points.
  This would help clarify the true added value of using LECOMH during testing, especially since LECOMH itself leverages all noisy annotations
  to determine consensus labels for training.

- The predictions of the human–AI selection model should incorporate the potential annotation costs associated with querying human annotators. One issue with implementing this consideration purely as a training-time regularization term is that it does not directly reflect the actual costs encountered during testing. Although increasing $\lambda$ leads to higher coverage, in practice it would be more useful to define a target coverage—such as a maximum budget for collaboration—and then determine the corresponding $\lambda$ value. Furthermore, the cost of requesting human input should be related to the cost of misclassification, for example, by applying decision-theoretic methods. Without these considerations, the practical applicability of LECOMH is limited.

- The authors differentiate between crowdsourced settings, where annotator accuracy varies widely, and multi-rater settings that rely on expert
  annotators. Although this distinction is clearly articulated, most of the references cited in Section 2.2:

  > In the inter-annotator approach, learning is performed to characterise inter-observer variabilities (Raykar et al., 2009; Guan et al., 2018;
  Mirikharaji et al., 2021; Ji et al., 2021), while in the intra-annotator approach, learning aims to estimate annotator-specific variabilities (e.g.,
  through confusion matrices) (Khetan et al., 2017; Tanno et al., 2019; Wu et al., 2022a; Cao et al., 2023).

  are majorly applied to datasets with crowdsourced labels. Accordingly, the methodological difference between learning from crowds and MRL remains unclear.

- It seems that no code has been provided for review, and it remains unclear whether the code will be made publicly available in the case of acceptance.

- The acknowledgments section includes a specific project name and corresponding ID. For a double-blind submission, such information should be removed beforehand.

- The authors refer to majority voting as a SOTA MRL technique in Section 5.1.3:

  > and SOTA MRL approaches, such as majority voting, UnionNet (Wei et al., 2022) and CROWDLAB (Goh et al., 2022).

  Yet, majority voting is rather a lower baseline in this area.

**Questions:**
- Why is the Gumbel-Softmax trick applied during testing to determine the number of annotators to be queried? The motivation for using it during training is clear, but during testing, wouldn’t it be sufficient to simply select the number of annotators with the highest probability? Or am I misunderstanding the concept in this context?
- How were the MLR techniques for comparison selected? There are many more (recent or popular) techniques [1, 2, 3, 4] dealing with annotations from multiple error-prone humans.

**References:**
- [1] Cao, Peng, et al. "Max-MIG: an Information Theoretic Approach for Joint Learning from Crowds." International Conference on Learning Representations. 2019.

- [2] Guo, Hui, Boyu Wang, and Grace Yi. "Label correction of crowdsourced noisy annotations with an instance-dependent noise transition model." Advances in Neural Information Processing Systems 36 (2023): 347-386.

- [3] Ibrahim, Shahana, Tri Nguyen, and Xiao Fu. "Deep Learning From Crowdsourced Labels: Coupled Cross-Entropy Minimization, Identifiability, and Regularization." The Eleventh International Conference on Learning Representations. 2023.

- [4] Zhang, Hansong, et al. "Coupled confusion correction: Learning from crowds with sparse annotations." Proceedings of the AAAI Conference on Artificial Intelligence. Vol. 38. No. 15. 2024.

---

> ### Author Response · Authors · 2025-01-28
> **Annotation Chosen and Clarification of CROWDLAB (W1)**
>
> We would like to thank the reviewer for this comment. Following the reviewer's suggestion, we have evaluated LNL trained with majority-vote labels for MRL. For the CIFAR-10N dataset, the LNL model trained with majority voting labels achieves an accuracy of 97.65\%, which is comparable to the 97.39\% accuracy reported in Table 2 for the model trained with random noisy labels. Furthermore, when integrated into LECOMH, it attains a peak accuracy of 98.84\% across all coverages, closely aligning with the results presented in Table 2, which reflect the performance of LNL trained with random noisy labels. However, for the Chaoyang dataset, the LNL model achieves 82.69\% accuracy which is comparable to the 82.44\% reported in Table 2. When integrated into LECOMH, it attains a peak accuracy of 99.27\% across all coverages, which are comparable to the results presented in our study in Table 2. Based on these findings, we comment about this majority voting setting in Section 5.1 for the LNL training, but will not include the results: LNL with random label or majority voting produce similar results, so we decided to use random label since it is simpler.
>
> For Section 5.3, we appreciate the reviewer's point about stronger baselines. To clarify, for the CROWDLAB baseline, we report results using the LNL models from Section 5.1 as the AI model, rather than the pre-trained vanilla ResNet18 with early stopping in Tables 1 and 2. This ensures consistency and a fairer comparison with LECOMH.

---

> > ### Comment · Reviewer_gDA4 · 2025-02-13
> >
> > Many thanks for the additional results confirming that LECOMH still performs superior in comparison to baselines using aggregated labels. However, I am confused by the result of LECOMH in **Table 1** for the dataset **IDN{20, 30, 50}**, which is $\mathbf{00.00 \pm 0.00}$. I assume that this is a placeholder for the actual results, which has not been inserted yet. Please insert the actual result in case of acceptance.

---

> > > ### Author Response · Authors · 2025-02-19
> > >
> > > Thanks for your reminder. We have updated the results in Table 1.

---

> ### Author Response · Authors · 2025-01-28
> **Training-time regularization term (W2)**
>
> Thank you for your insightful feedback. We now discuss the  limitation about the cost of misclassification in Section 6, as follows: "One excessive simplification of our method is that it assumes the cost of a user input to be a flat value of "1", representing the initial step in formulating HAI-CC methods. However, more sophisticated cost models need to be developed to account for the impacts of false positives and false negatives, particularly in the context of patient-specific conditions (e.g., the high cost of a false negative). We are actively collaborating with health economists to study these factors and plan to incorporate such nuanced cost models in future iterations of this work."
>
> Regarding the target coverage issue, it is worth noting that current HAI-CC methods predominantly rely on post-hoc techniques to generate accuracy-coverage curves. These approaches involve sorting deferral scores and adjusting prediction thresholds to achieve the desired coverage level. However, this post-hoc approach has two significant drawbacks: (1) it requires access to the entire test set prior to making predictions, and (2) it lacks a mechanism to enforce or evaluate workload control during the training process.
> In contrast, our method addresses these limitations by introducing $\lambda$ as a hyper-parameter that can be directly mapped to a specific coverage value. This enables proactive control over coverage during training, eliminating the need for post-hoc adjustments.
> As a future work, we aim to develop optimisation strategies that explicitly constrain models to achieve particular coverage values, as suggested by the reviewer. This will provide a more robust and principled framework for analysing and visualising accuracy-coverage trade-offs.

---

> ### Author Response · Authors · 2025-01-28
> **Difference between learning from crowds and MRL (W3)**
>
> We appreciate the reviewer's observation regarding the distinction between learning from crowds (LFC) and multi-rater learning (MRL) settings. To address this, we have renamed and expanded Section 2.2 to include a more detailed discussion of the methodological differences between LFC and MRL. Specifically, we clarify how MRL focuses on leveraging the high consistency and expertise of annotators to model inter- and intra-annotator variability, in contrast to LFC settings, where annotator accuracy tends to vary widely.

---

> > ### Comment · Reviewer_gDA4 · 2025-02-13
> >
> > The detailed Section 2.2 mentions many important developments and approaches in the **LFC and MRL setting**. However, there are still a few issues from my side that could be addressed in the case of acceptance:
> >
> > > ... LFC algorithms ... may be similar or even interchangeable with MRL methods, the objectives ... differ significantly
> >
> > I would say both types of approaches have the same goal, i.e., achieving maximum performances, but differ in their
> > assumptions.  Moreover, looking at the references in the Sections 2.2.1 and 2.2.2, I cannot recognize how you made this
> > distinction. For example, you reference the works [1, 2] as LFC approaches, while you declare [3, 4] as MLR approaches. However, the works [1]  and [3] only differ in a regularization term as part of their loss function. Moreover, the authors of [1] even compare their work to [2]. A similar observation can be made for the comparison of the works [2] and [4]. Both works use noise adaption layers (with slight architectural changes regarding the modelling of annotator interdependencies) and the authors of [4] also compare their results to the ones of the approach in [2].
> >
> > > ... while in the intra-annotator strategy, learning aims to estimate annotator-specific variabilities (e.g., through confusion
> > matrices) (Khetan et al., 2017; Tanno et al., 2019; Wu et al., 2022a; Cao et al., 2023).  A joint learning method combining both the inter- and intra-annotator strategies was proposed to integrate the strengths of both approaches (Wu et al., 2022a). One drawback in those studies is the assumption of sample-independence, ...
> >
> > To which works does the term "studies" refer? It seems that studies also refer to the work of Cao et al [5]. However, this work does not share the mentioned drawback, but models sample-dependent reliabilities of annotations by combining annotator expertise and instance difficulty.
> >
> > **References:**
> > - [1] Ibrahim, Shahana, Tri Nguyen, and Xiao Fu. "Deep Learning From Crowdsourced Labels: Coupled Cross-Entropy Minimization, Identifiability, and Regularization." The Eleventh International Conference on Learning Representations. 2023.
> > - [2] Filipe Rodrigues and Francisco Pereira. Deep Learning from Crowds. In AAAI Conference on Artificial
> > Intelligence, volume 32(1), 2018.
> > - [3] Ryutaro Tanno, Ardavan Saeedi, Swami Sankaranarayanan, Daniel C Alexander, and Nathan Silberman.
> > Learning from Noisy Labels by Regularized Estimation of Annotator Confusion. In IEEE Conference on
> > Computer Vision and Pattern Recognition, pp. 11244–11253, 2019.
> > - [4] Hongxin Wei, Renchunzi Xie, Lei Feng, Bo Han, and Bo An. Deep Learning from Multiple Noisy Annotators
> > as a Union. IEEE Transactions on Neural Networks and Learning Systems, 2022.

---

> > > ### Author Response · Authors · 2025-02-19
> > >
> > > We thank the reviewer for the comment. We acknowledge the confusion in SubSection 2.2 when discussing label aggregation methods. Since our paper focuses on human-AI collaboration, we simplify this subsection by discussing different multi-rater learning methods to obtain consensus labels. In the revision, the changes are highlighted in red.

---

> ### Author Response · Authors · 2025-01-28
> **Code of LECOMH (W4)**
>
> We provide the code in the supplemental materials and will publicly release the code after acceptance.

---

> ### Author Response · Authors · 2025-01-28
> **Acknowledgments section (W5)**
>
> We apologise for that and remove the section.

---

> ### Author Response · Authors · 2025-01-28
> **Strong MRL baselines (W6 and Q2)**
>
> Thank you for this feedback and we apologise for citing majority voting as SOTA. We meant to cite majority voting as a baseline and we have included additional results in Tables 1 and 2, incorporating the references [1, 2, 3, 4] suggested by the reviewer. These additions provide a more comprehensive comparison of MRL techniques. We hope these updates can clarify the context of our method.

---

> > ### Comment · Reviewer_gDA4 · 2025-02-13
> >
> > Great to see the additional results for the **MRL/LFC approaches**, confirming the performance gains of LECOMH.
> >
> > As a very minor remark: Could you specify in the text, which variants of UnionNet (A vs. B) and GeoCrowdNet (F vs. W) did you use?

---

> > > ### Author Response · Authors · 2025-02-19
> > >
> > > We used UnionNet B and GeoCrowdNet F. We modified the text in Section 5.3 and Tables 1 and 2 to explicitly mention them with their variants.

---

> ### Author Response · Authors · 2025-01-28
> **Gumbel-Softmax trick during testing (Q1)**
>
> Thank you very much for this important question. This is indeed a limitation of LECOMH that is now discussed in Section 6. The challenge lies in the behaviour of the Human-AI Selection Module when determining the number of users to collaborate with the AI classifier. If we apply an argmax operation on the module's output, the result almost always favours selecting the AI model alone for predictions. However, by using Gumbel-Softmax sampling from the categorical distribution of the module's output, we can achieve a more balanced and accurate classification. This issue is illustrated in Tables 3 and 4: while the AI model consistently has the highest probability of being selected, the sampling mechanism occasionally forces collaboration between the AI model and the users, ensuring greater diversity and accuracy in the decision-making process.

---

> > ### Comment · Reviewer_gDA4 · 2025-02-13
> >
> > Thank you for this helpful explanation. Including a brief sentence in Section 3.2 that explains the motivation behind using the Gumbel-Softmax trick during testing would clarify your explained rationale earlier.

---

> > > ### Author Response · Authors · 2025-02-19
> > >
> > > We thank the reviewer for the suggestion. We have added an explanation in Section 3.2 to clarify further our motivation when using Gumbel-Softmax trick in red colour. The added text is: "Note that the human-AI selection module’s behavior poses a challenge in determining the number of users to collaborate with the AI classifier, an argmax operation on the module’s output tends to favor the AI model alone.".

---

> ### Comment · Reviewer_gDA4 · 2025-02-13
>
> The clarifications regarding the **annotation cost-related aspects** of LECOMH and pointing out in your conclusion that there is still work to be done in this regard improved your paper. However, I am still confused by the following statement in your reply:
>
> > ... by introducing $\lambda$ as a hyper-parameter that can be directly mapped to a specific coverage value ...
>
> Fig. 6 (b) indicates different coverage values across datasets for the same $\lambda$ value. So, I rather assume that you mean that by tweaking the value of $\lambda$ per dataset, you can approximately achieve a certain coverage.
>
> **Off-topic:** In the corresponding description of Fig. 6 (added blue text at the end Section 5.4), you refer to "Eq.5" and "Eq.4" missing compared to the reference "Eq. (4)" in the caption of Fig. 6. For such references and others, a consistent notation would be nice in the case of acceptance.

---

> > ### Author Response · Authors · 2025-02-19
> >
> > As mentioned at the end of the first paragraph in Section 5.2, the hyper-parameter \(\lambda\) value is adjusted to achieve certain level of coverage (or cost). This has been done to plot the results shown in Figs. 4 - 6.
> >
> > For the inconsistent notation in Fig. 6, we have updated the notation in the text to make it consistent. Thank you, the reviewer, for this.

---

### Review · Reviewer_7oMd · 2024-12-29

**Summary Of Contributions:**

The paper proposes the LECOMH method for human-AI collaborative classification (HAI-CC), enabling training exclusively on noisy labels while optimizing collaborative accuracy and minimizing human collaboration costs. It also introduces new benchmarks to evaluate HAI-CC methods on datasets with multiple noisy labels in both training and testing phases.

**Audience:**

Yes

**Claims And Evidence:**

Yes

**Requested Changes:**

1. Clearly differentiate LECOMH from prior works, particularly in terms of algorithmic innovation beyond the integration of existing components.


2. Expand experiments to systematically analyze the cost-accuracy trade-offs across different noise levels


3. Introduce experiments that isolate the contributions of LNL and MRL components to validate their individual effectiveness.


4. Add experiments on real-world noisy datasets to better reflect the variability and challenges encountered in practical scenarios

**Strengths And Weaknesses:**

**Strengths:**
1. Practical Motivation: Addresses real-world scenarios where clean labels are unavailable, making the approach potentially practical.


2. Cost Optimization: Focuses on minimizing human collaboration costs, an important consideration for scalability.

**Weakness:**
1. While the integration of LNL and MRL is interesting, it primarily builds on existing methods (e.g., CROWDLAB) without introducing fundamentally new algorithms. The two-stage method lacks novelty.


2. Assumptions that all annotators have similar performance levels, which is often unrealistic.


3. The method emphasizes cost minimization but lacks sufficient exploration of the trade-off with accuracy, particularly in high-noise scenarios. Figure 4 demonstrates the trade-off between accuracy and coverage by varying the cost parameter ($\lambda$), but it does not provide a detailed analysis for different noise levels.



4. The reliance on synthetic or consensus-based labels (e.g., CIFAR-10N and Chaoyang) may not adequately represent the variability and challenges of real-world noisy datasets, limiting the generalizability of the results.

---

> ### Author Response · Authors · 2025-01-28
> **Differentiate LECOMH from prior works (W1,RC1)**
>
> LECOMH is the first HAI-CC framework capable of being trained on real-world datasets with multiple noisy labels per image. In contrast, all other HAI-CC methods known to us require clean labels for training, which represents a critical distinction from previous approaches.
> We addressed this challenge by leveraging learning with noisy labels (LNL) and multi-rater learning (MRL) techniques, enabling LECOMH to train effectively on noisy-label datasets without relying on clean training labels.
> Another significant contribution of LECOMH is its introduction of a novel learning to complement with multiple users approach. Our method optimises both HAI-CC classification accuracy and collaboration cost simultaneously, which consists of an innovative optimisation, as prior learning-to-complement methods do not address these two objectives jointly.
> These contributions required us to introduce a new training algorithm and a new optimisation loss function.

---

> ### Author Response · Authors · 2025-01-28
> **Experiments on different noise levels (W2, W3, RC2)**
>
> We would like to thank the reviewer for this question. In our original experiments on CIFAR10 dataset, the noise levels for the annotators were homogeneous, but for Chaoyang and NIH datasets, the noise levels for the annotators are quite different. In Section 4.3 of our revision, we now explain that “The prediction accuracy of the 3 users in the NIH-AO dataset is approximately 89\%, 94\%, 80\% in training and 89\%, 94\%, 80\% in testing.” For the Chaoyang dataset, the three pathologists also have different noise levels, which is now explained with the following sentence in Section 4.2 of our revision: "The prediction accuracy of the three users in the dataset is approximately 93\%, 88\%, 99\% in training and 88.7\%, 86.9\%, 99\% in testing."
>
> To clarify this issue further, we simulate three experts with different noise levels (with IDN noise rates at 20\%, 30\% and 50\%, denoted by IDN\{20, 30, 50\}) to evaluate the performance of LECOMH and other methods in such condition.  Results are shown in Tables 1 and 2, and Figure 8. The results show that LECOMH is more accurate than other methods in this experiment. We added a discussion about this experiment in Section 5.3.2.

---

> ### Author Response · Authors · 2025-01-28
> **Isolate the contributions of LNL and MRL components (RC3)**
>
> *Introduce experiments that isolate the contributions of LNL and MRL components to validate their individual effectiveness.*
>
> We are reporting the accuracy of LNL and MRL in Tables 1 and 2. Also, in the ablation study (section 5.3), we are showing the performance of our method without LNL and without MRL in Table 5.

---

> ### Author Response · Authors · 2025-01-28
> **Experiments on other real-world noisy datasets (RC4)**
>
> *Add experiments on real-world noisy datasets to better reflect the variability and challenges encountered in practical scenarios*
>
> Please note that the experiments using CIFAR-10H, Chaoyang and NIH represent real-world multi-noisy label datasets. We are not aware of other publicly available real-world datasets that contain multiple noisy labels provided by humans. We are happy to evaluate our method on additional real-world datasets that the reviewer can recommend.

---

> ### Comment · Reviewer_gDA4 · 2025-02-13
>
> For future work or a potential camera-ready version in the case of acceptance, I'd like to inform the authors about additional datasets (work [1] already cited by you) containing instances with multiple noisy labels from humans, which are:
>
> - label-me [1] as an image classification dataset with mostly about 2 to 3 noisy labels per instance,
> - sentiment polarity [2] as a text classification dataset with mostly about 5 to 6 noisy labels per instance,
> - music genres [2] as an audio classification dataset with mostly about 4 to 5 noisy labels per instance,
> - reuters [3] as a text classification dataset with mostly about 3 noisy labels per instance,
> - a collection of datasets (e.g., benthic, mice bone, pig, plankton, etc.) [4] with quite a few (depending on the dataset, between 4 and 100) labels per instance,
> - dopanim [5] as an image classification dataset with mostly about 5 noisy labels per instance.
>
> Some of these datasets only contain multiple noisy labels from humans for training instances. In such cases, one would need to create new training and test splits.
>
> **References:**
> - [1] Rodrigues, F. and Pereira, F. Deep Learning from Crowds. In AAAI Conference on Artificial Intelligence, volume 32(1), 2018.
> - [2] Rodrigues, F., Pereira, F., and Ribeiro, B. Learning from multiple annotators: distinguishing good from random labelers. Pattern Recognition Letters, pp. 1428–1436, 2013.
> - [3] Rodrigues, F., Lourenço, M, Ribeiro, B, Pereira, F. Learning Supervised Topic Models for Classification and Regression from Crowds. IEEE Transactions on Pattern Analysis and Machine Intelligence (TPAMI), 2017.
> - [4] Schmarje, L., Grossmann, V., Zelenka, C., Dippel, S., Kiko, R., Oszust, M., Pastell, M., Stracke, J., Valros, A., Volkmann, N. and Koch, R., 2022. Is one annotation enough?-a data-centric image classification benchmark for noisy and ambiguous label estimation. Advances in Neural Information Processing Systems, 35, pp.33215-33232.
> - [5] Herde, M., Huseljic, D., Rauch, L. and Sick, B., dopanim: A Dataset of Doppelganger Animals with Noisy Annotations from Multiple Humans. In The Thirty-eight Conference on Neural Information Processing Systems Datasets and Benchmarks Track, 2024.

---

> > ### Author Response · Authors · 2025-02-19
> >
> > We thank the reviewer for the suggestion of the extensive list of datasets available. Among all the suggested datasets, some are applicable for our evaluation due to their size and the available of multiple annotations per sample. We will include them into our future works, especially Sentiment Polarity [2], MiceBone [4] and dopanim [5]. For the current paper, we also include the results evaluated on the MiceBone dataset and show in Appendix A.

---

### Review · Reviewer_gwQ5 · 2025-01-16

**Summary Of Contributions:**

This paper introduces LECOMH, a human-AI collaborative classification (HAI-CC) method that learns exclusively from multiple *noisy* labels (i.e. without requiring clean training data) and collaborates with multiple experts. At its core is a three-component architecture that integrates LNL pre-training, multi-rater learning via CROWDLAB, and a Gumbel-softmax based selection module for optimizing both accuracy and human collaboration costs.

The experiments demonstrate consistent improvements over SOTA methods across multiple datasets:
1. CIFAR-10H/N: 98.82% vs 97.76% (CET baseline) at 50% coverage
2. Chaoyang medical imaging: 99.79% vs 99.20% (CET)
3. NIH chest X-ray datasets: 94.85% vs 94.14% (CET) for airspace opacity detection

Additionally, the paper introduces new benchmarks for evaluating human-AI collaborative systems with multiple noisy labels in both training and testing, providing more realistic evaluation settings.

**Audience:**

Yes

**Broader Impact Concerns:**

The paper should expand its discussion of ethical implications, particularly:
1. Expert de-skilling risks in the medical domain,
2. Potential amplification of annotation biases,
3. Privacy considerations for expert annotation collection

**Claims And Evidence:**

Yes

**Requested Changes:**

1. More theoretical analysis, e.g. analyzing how noise levels affect consensus label quality; or bounds on the accuracy-cost trade-off.
2. Address expert heterogeneity through extension to varying expert reliability. Maybe also include comparison with methods that model annotator reliability.

**Strengths And Weaknesses:**

# Strengths

## 1. Key Technical Contributions:
The method successfully bridges an important gap between LNL/MRL methods and human-AI collaboration through
1. Integration of SOTA LNL pre-training (ProMix, InstanceGM) with CROWDLAB for consensus generation, showing strong performance across noise levels (20-50% on IDN datasets);
2. Novel use of Gumbel-softmax for differentiable collaboration selection, enabling efficient end-to-end training

## 2. Comprehensive Evaluation:
The experimental validation is thorough and convincing. Ablation studies in Table 5 clearly demonstrate each component's contribution. The method shows strong performance on both clean labels (98.82% CIFAR-10H) and noisy scenarios (96.08% IDN50)


# Weaknesses
## 1. Expert Modeling
The main limitation of the method is that there is no explicit handling of varying expert reliability as seen in real medical scenarios. It assumes homogeneous expert performance despite evidence of varying reliability (Table 1: 79.36%-49.30% accuracy range on IDN)

## 2. Theoretical analysis
The paper could benefit from some analysis of formal convergence guarantees for the optimization and bounds on collaboration cost vs accuracy trade-offs.

## 3. Dataset coverage
This work only explore image classification task. Can this problem be studied under different problem statements?

---

> ### Author Response · Authors · 2025-01-28
> **Convergence guarantees for the optimization (W2)**
>
> The proposed method achieves convergence by leveraging the well-established principles of stochastic gradient descent (SGD) [A], whose convergence properties have been extensively validated in prior research. As the convergence analysis of SGD is not a novel contribution of this work, we have chosen to omit it from the paper.
>
> [A]Gower R M, Loizou N, Qian X, et al. SGD: General analysis and improved rates[C]//International conference on machine learning. PMLR, 2019: 5200-5209.

---

> ### Author Response · Authors · 2025-01-28
> **Theoretical analysis on accuracy-cost bounds and noise levels affect consensus label quality (RC1,W2)**
>
> We appreciate this question by the reviewer. We have added a theoretical analysis of how  noise levels affect consensus label quality in Section 3.3.

---

> ### Author Response · Authors · 2025-01-28
> **Expert heterogeneity (RC2,W1)**
>
> In our original experiments on CIFAR10 dataset, the noise levels for the annotators were homogeneous, but for Chaoyang and NIH datasets, the noise levels for the annotators are quite different. In Section 4.3 of our revision, we provide further details on the performance of each human experts in NIH-AO dataset as follows: "The prediction accuracy of the 3 users in the NIH-AO dataset is approximately 89\%, 94\%, 80\% in training and 89\%, 94\%, 80\% in testing." For the Chaoyang dataset, the three pathologists also have different noise levels. To increase clarity, we provide additional details of their performance in Section 4.2 of our revision as follows: "The prediction accuracy of the three users in the dataset is approximately 93\%, 88\%, 99\% in training and 88.7\%, 86.9\%, 99\% in testing."
>
> We also simulate three synthetic experts with different noise levels (with IDN noise rates at 20\%, 30\% and 50\%, denoted by IDN\{20, 30, 50\}) to evaluate the performance of LECOMH and other methods in such condition. The results in Tables 1 and 2, and Figure 8 show that LECOMH is more accurate than other methods in this experiment. We add a discussion about this experiment in Section 5.3.2 of our revised paper.
>
> In Tables 1 and 2, we compare LECOMH with  SOTA MRL methods that model annotator reliability (Max-MIG[1], BayesianIDNT[2], GeoCrowdNet[3], and CCC[4])).
>
> [1] Cao, Peng, Yilun Xu, Yuqing Kong, and Yizhou Wang. "Max-MIG: An information theoretic approach for joint learning from crowds." International Conference on Learning Representations. 2019.
>
> [2] Guo, Hui, Boyu Wang, and Grace Yi. "Label correction of crowdsourced noisy annotations with an instance-dependent noise transition model." Advances in Neural Information Processing Systems 36 (2023): 347-386.
>
> [3] Ibrahim, Shahana, Tri Nguyen, and Xiao Fu. "Deep Learning From Crowdsourced Labels: Coupled Cross-Entropy Minimization, Identifiability, and Regularization." International Conference on Learning Representations. 2023.
>
> [4] Zhang, Hansong, et al. "Coupled confusion correction: Learning from crowds with sparse annotations." Proceedings of the AAAI Conference on Artificial Intelligence. Vol. 38. No. 15. 2024.

---

> ### Author Response · Authors · 2025-01-28
> **Different problem statements (W3)**
>
> The only non-image classification dataset with multiple noisy labels per sample that we are aware of is the HateSpeech dataset [5]. However, since this dataset is synthesized, we chose not to include it in this submission.
>
> [5] Davidson T, Warmsley D, Macy M, et al. Automated hate speech detection and the problem of offensive language[C]//Proceedings of the international AAAI conference on web and social media. 2017, 11(1): 512-515.

---

> ### Author Response · Authors · 2025-01-28
> **Discussion of Expert de-skilling risks in the medical domain, Potential amplification of annotation biases, Privacy considerations for expert annotation collection**
>
> That is indeed an important point to raise in this paper. We have expanded the discussion in Section 6 to highlight that LECOMH may contribute to expert de-skilling if human experts collaborating with the system become overly reliant on its predictions, potentially leading to overconfidence and a decline in their independent decision-making skills.

---

### Review · Reviewer_T1TB · 2025-02-15

**Summary Of Contributions:**

This paper proposes the Learning to Complement with Multiple Humans (LECOMH) approach to address the issue of learning from noisy labels in human-AI collaborative classification. It first reviews related work on noisy-label learning, multi-rater learning, and human-AI collaborative classification. Then, it details the LECOMH method, including its training and testing procedures and a theoretical analysis of label noise. The authors also introduce new benchmarks for evaluation. Experimental results show that LECOMH outperforms existing methods in terms of accuracy and cost. However, the method has limitations such as assuming similar labeller performance and relying on sampling in testing. Overall, it presents a novel solution with potential but also areas for improvement.

**Audience:**

Yes

**Claims And Evidence:**

Yes

**Requested Changes:**

See Weaknesses.

**Strengths And Weaknesses:**

Strengths：

1. The proposed LECOMH method is a novel attempt to integrate multiple noisy labels in human - AI collaborative classification. It fills the research gap between traditional noisy - label learning methods that don't collaborate with users during testing and existing HAI - CC methods that rely on clean labels. This innovative idea has the potential to significantly improve the performance of human - AI collaborative systems in real - world scenarios where noisy labels are prevalent.

2. The introduction of new benchmarks for HAI - CC methods, such as the ones based on CIFAR - 10, Chaoyang, and NIH datasets, is a valuable contribution. These benchmarks, which include multiple noisy labels in both training and testing, provide a more realistic and comprehensive evaluation environment for comparing different methods. They can serve as a standard for future research in this field.

3.  The paper conducts extensive and well - designed experiments. It compares LECOMH with a wide range of baseline methods, including SOTA HAI - CC, LNL, and MRL methods. The evaluation metrics, such as accuracy - coverage curves and accuracy - cost curves, provide a comprehensive view of the method's performance. The ablation studies also help to understand the contribution of each component of LECOMH.


Weaknesses:

1. LECOMH assumes that all labellers have similar performance, which is a significant simplification. In real - world scenarios, labellers' accuracies can vary greatly. This assumption may limit the method's performance when applied to datasets with highly heterogeneous labellers. It would be better if the method could adapt to different labeller performances without this strong assumption.

2. The cost model in LECOMH is overly simplistic, treating the cost of each user input as a flat value. In practice, the cost of false positives and false negatives can vary significantly, especially in applications like medical diagnosis. A more sophisticated cost model that takes these factors into account is needed to make the method more practical and accurate.

3. Although the paper conducts experiments on multiple datasets, the generalization ability of LECOMH to other types of datasets or real - world scenarios is not fully explored. It's unclear whether the method can perform well on datasets with different characteristics, such as different levels of noise, different data distributions, or different types of tasks.

---

> ### Author Response · Authors · 2025-02-19
> **Expert heterogeneity and Real-world Dataset (W1 and W3)**
>
> On one hand, we agree with the reviewer about the homogeneous noise in synthetic datasets (e.g., CIFAR-10).
> We also simulate three synthetic experts with different noise levels (with IDN noise rates at 20\%, 30\% and 50\%, denoted by IDN\{20, 30, 50\}) to evaluate the performance of LECOMH and other methods in such condition. The results in Tables 1 and 2, and Figure 8 show that LECOMH is more accurate than other methods in this experiment. We add a discussion about this experiment in Section 5.3.2 of our revised paper.
>
> On the other hand, we disagree in the cases of real-world datasets, such as Chaoyang and NIH. In those datasets, the characteristic of each labeller is not the same. Specifically,
>
>  - Chaoyang: The prediction accuracy of the three annotators is approximately 93\%, 88\%, 99\% in training and 88.7\%, 86.9\%, 99\% in testing,
>  - NIH-AO: The prediction accuracy of the 3 labellers is approximately 89\%, 94\%, 80\% in training and 89\%, 94\%, 80\% in testing.
>
>  We also update Section 4.2 and 4.3 to reflect such information about these two real-world datasets.
>
> Furthermore, we also conduct experiments on Micebone, which is a real-world dataset. The accuracy of 8 experts are from 84\% to 86\%. The results in Figure 10 in the Appendix show that LECOMH also performs better than other methods in this experiment.

---

> ### Author Response · Authors · 2025-02-19
> **Discussion of Cost Model**
>
> Thank you for your insightful feedback. We now discuss the limitation about the cost of misclassification in Section 6, as follows: "One excessive simplification of our method is that it assumes the cost of a user input to be a flat value of "1", representing the initial step in formulating HAI-CC methods. However, more sophisticated cost models need to be developed to account for the impacts of false positives and false negatives, particularly in the context of patient-specific conditions (e.g., the high cost of a false negative). We are actively collaborating with health economists to study these factors and plan to incorporate such nuanced cost models in future iterations of this work."

---

### Decision · Action_Editor_skJa · 2025-02-26

**Recommendation:** Reject

**Comment:**

This article proposes an improved variant of the HAI-CC class of methods. Although the authors have demonstrated through extensive experiments that this method can consistently enhance the accuracy of models under noisy label learning conditions, there are two notable shortcomings in the article.

Firstly, the method lacks validation results on real-world datasets. Almost all reviewers have pointed out the significance of this issue. Regrettably, the authors did not include additional experiments in their rebuttal.

Secondly, there are still questions regarding the interpretability of the method's effectiveness. For instance, the selection of the hyperparameter lambda primarily relies on experimentation, which further hinders the practicality of the method. Also, reviewers have also questioned the assumption made in this paper, which is not properly supported with empirical or theoretical evidence.

Taking these factors into account, I have decided to reject this paper.

**Audience:**

This article will be of interest to  researchers who work on Noisy Label Learning, especially for those who focus on the HAI-CC methodology.

**Claims And Evidence:**

This paper claims  to propose a novel Learning to Complement with Multiple Humans (LECOMH) approach, which addresses the gap between HAI-CC method and real-world application. The authors provide an extensive empirical evaluation to support this claim. On one hand, the improved quantitative performance shows the effectiveness of the proposed method. On the other hand, the evaluated benchmarks are mostly synthetic. As pointed out by the reviewers, the author did not prove their effectiveness on real-world dataset.

**Resubmission Of Major Revision:**

The authors may consider submitting a major revision at a later time.